# Discovery of four *Noggin* genes in lampreys suggests two rounds of ancient genome duplication

Galina V. Ermakova[1], Alexander V. Kucheryavyy[2], Andrey G. Zaraisky[1✉] & Andrey V. Bayramov [1✉]

The secreted protein Noggin1 was the first discovered natural embryonic inducer produced by cells of the Spemann organizer. Thereafter, it was shown that vertebrates have a whole family of *Noggin* genes with different expression patterns and functional properties. For example, Noggin1 and Noggin2 inhibit the activity of BMP, Nodal/Activin and Wnt-beta-catenin signalling, while Noggin4 cannot suppress BMP but specifically modulates Wnt signalling. In this work, we described and investigated phylogeny and expression patterns of four *Noggin* genes in lampreys, which represent the most basally divergent group of extant vertebrates, the cyclostomes, belonging to the superclass Agnatha. Assuming that lampreys have *Noggin* homologues in all representatives of another superclass of vertebrates, the Gnathostomata, we propose a model for *Noggin* family evolution in vertebrates. This model is in agreement with the hypotheses suggesting two rounds of genome duplication in the ancestor of vertebrates before the divergence of Agnatha and Gnathostomata.

[1] Severtsov Institute of Ecology and Evolution, Russian Academy of Sciences, Moscow 119071, Russia. [2] Shemyakin-Ovchinnikov Institute of Bioorganic Chemistry, Russian Academy of Sciences, Moscow 117997, Russia. ✉email: azaraisky@yahoo.com; andrbayr@gmail.com

The divergence of Agnatha (jawless fishes) and Gnathostomata (gnathostomes) occurred ~535–462 MYA, i.e., at the earliest stages of the evolution of vertebrates in the Palaeozoic Era[1–3]. These data allow one to consider jawless fishes, including extant lampreys and hagfishes, as the most basally divergent group of vertebrates. In turn, this early separation of cyclostomes from other vertebrates makes them a recognized model for studying the genetic innovations that led to the evolutionary emergence and development of unique traits of the vertebrates, such as the telencephalon, neural crest cells, the epimorphic regeneration ability of the body appendages, and adaptive immunity[4–6]. It is generally accepted that the genetic basis of such innovations is genomic duplications; however, the number and timing of these duplications still need to be clarified[7,8]. One possible approach to resolve this issue involves comparing the evolution, expression and functions of genes that regulate basic processes of vertebrate development in representatives of cyclostomes and other vertebrates. In the present work, we performed such analysis for the genes of the *Noggin* family, which encode secreted factors participating in the regulation of the development of many organs during vertebrate embryogenesis[9–16].

For a long time, it was believed that vertebrates have only one "classical" *Noggin* gene, which functions in vertebrate development by suppressing the BMP (bone morphogenetic protein) signalling pathway by sequestering BMP ligands[9,10]. However, first in amphibians and then in other gnathostomes, homologues of the "classical" *Noggin*, namely, *Noggin2* and *Noggin4* were described[17,18]. Moreover, it was shown that the role of these Noggin proteins in the development of vertebrates was not limited to the previously described inhibition of the BMP signalling pathway but also includes suppression of Nodal/Activin and Wnt signalling[15,16,19,20]. Thus, as modulators of these three basic intracellular signalling pathways, Noggin proteins can play important roles in many processes of early tissue differentiation.

The presence of several *Noggin* genes also in cyclostomes was noticed for the first time for the sea lamprey *Petromyzon marinus*[21]. However, since this work was focused on the genome-wide analysis of genes lost in the early mammalian evolution, these lamprey *Noggins*, while included in common phylogenetic analysis, were not described and investigated in detail. In the present work, we revealed that lampreys have four *Noggin* genes and then studied their phylogeny, local genomic synteny, expression patterns and the ability to induce secondary body axes, including the head. Establishing that lampreys have orthologues of *Noggin1*, *Noggin2* and *Noggin4* in gnathostomes, we consider their possible evolution in the context of the existing models of genome duplications in the early vertebrate history. The obtained results incline us to the hypothesis suggesting two rounds of genome duplication in the common ancestor of vertebrates before divergence of jawless and gnathostomes.

## Results

**Phylogenetic analysis of lamprey *Noggin* genes**. The available lamprey genome databases (whole-genome shotgun contigs of the Arctic lamprey, *Lethenteron camtschaticum*, and sea lamprey, *P. marinus*) were used to search for putative lamprey homologues of known *Noggin* family genes. We were able to simplify the screening because the lamprey *Noggin* coding sequences presumably should not contain introns, since the absence of introns was previously shown for all known *Noggin* genes in both invertebrates and vertebrates. As a result, we revealed in the *L. camtschaticum* and the *P. marinus* genomes the following four genes, which indeed had no introns, with protein products that demonstrated homology with the Noggin1, Noggin2 and Noggin4 described in gnathostomes[9,17,18]:

*NogginA* (*L. camtschaticum* contig 015835, sequence ID: APJL01043027.1; *P. marinus* isolate animal number 11 contig 46719, Sequence ID: AEFG01046720.1);

*NogginB* (*L. camtschaticum* contig 045160, sequence ID: APJL01075429.1; *P. marinus* isolate animal number 11 contig 47771, sequence ID: AEFG01047772.1);

*NogginC* (*L. camtschaticum* unplaced genomic scaffold scaffold00115, sequence ID: KE993786.1; *P. marinus* isolate animal number 11 contig 36440, sequence ID: AEFG01036441.1); and

*NogginD* (*L. camtschaticum* contig 002783, sequence ID: APJL01035689.1; *P. marinus* isolate animal number 11 contig 9676, sequence ID: AEFG01009677.1).

Based on these sequences, we designed primers and cloned the full-length cDNAs of four *Noggin* genes of European river lamprey (*Lampetra fluviatilis*) *Noggin* genes. According to the available literature and our own data, this species is extremely close to *L. camtschaticum* in terms of genomic sequences and developmental traits, while its embryos are much more accessible, at least in our circumstances than those of the latter.

The alignment of the protein sequences encoded by these genes showed that all of them have conserved cysteine residues known to be important for the formation of functional Noggin dimers[16,22] (Supplementary Fig. 1).

As we also revealed, by maximum likelihood (ML) protein analysis, NogginA appeared to be closer to gnathostome Noggin1, while NogginB and NogginC appeared to be closer to Noggin2. However, as the bootstrap value justifying such clustering is quite low (<50%) in our opinion, it would be more correct to speak of a cluster or cloud homology of the lamprey NogginA/B/C proteins on the one hand and jawed vertebrates' Noggin1/2 on the other. (Fig. 1a). Adding of *Echinodermata* Noggin genes to the tree or using the Neighbor-joining (NJ) algorithm doesn't clear up the picture (Supplementary Fig. 2). At the same time, NogginD was confidently grouped with Noggin4. Moreover, NogginD, the lamprey orthologue of Noggin4, had amino acid substitutions in positions critical for the binding of BMP, which presumably would prevent its binding with these ligands (Supplementary Fig. 1)[16,22].

Hoping to clarify the phylogenetic relationship of lamprey and jawed vertebrates' *Noggin* genes, we analysed their local genomic synteny in the genomes of the lamprey *P. marinus* and the frog *Xenopus tropicalis*. As a result, we found that *NogginA* of *P. marinus* and *Noggin1* of *X. tropicalis* both have at least two common genes flanking their 3′ end, and in the *X. tropicalis* genome, they are designated *C17ORF67* (ENSXETG00000007759) and *synaptogyrin2* (*Syngr2*, ENSXETG00000007946). This finding indicates that both of these *Noggin* genes could derive from a common ancestor, as was predicted by the clustering analysis.

At the same time, *NogginB* and *NogginC* also have neighbouring genes in common with *X. tropicalis Noggin1*, but flanking the 5′ end. Interestingly, although *NogginC* has two gene homologues of *X. tropicalis ANKFN* (XM_018097804) and *MMD* (ENSXETG00000007598), *NogginB* adjoins with only *ANKFN*. Notably, the paralogue of the latter, *ANKFN1L* (XM_002932447.4), was found near the 5' end of *X. tropicalis Noggin2*. Taken together, these results suggest that *NogginB* and *NogginC* of *P. marinus*, as well as *Noggin2* of *X. tropicalis*, likely have a common ancestor with *Noggin1* and *NogginA* but they probably form another branch of genes derived from this ancestor. This conclusion is consistent with the results of a clustering analysis that slightly gravitates *NogginA* to *Noggin1* in one cluster and *NogginB*, *NogginC* to *Noggin2* in another cluster.

In contrast to *NogginA*, *B* and *C*, lamprey *NogginD* has no proximal genes homologous to the genes adjacent to *Noggin1* in *X. tropicalis*. However, at least two genes, homologous to *nubp2* (ENSXETG00000010354) and *KCNJ4* (ENSXETG00000010341)

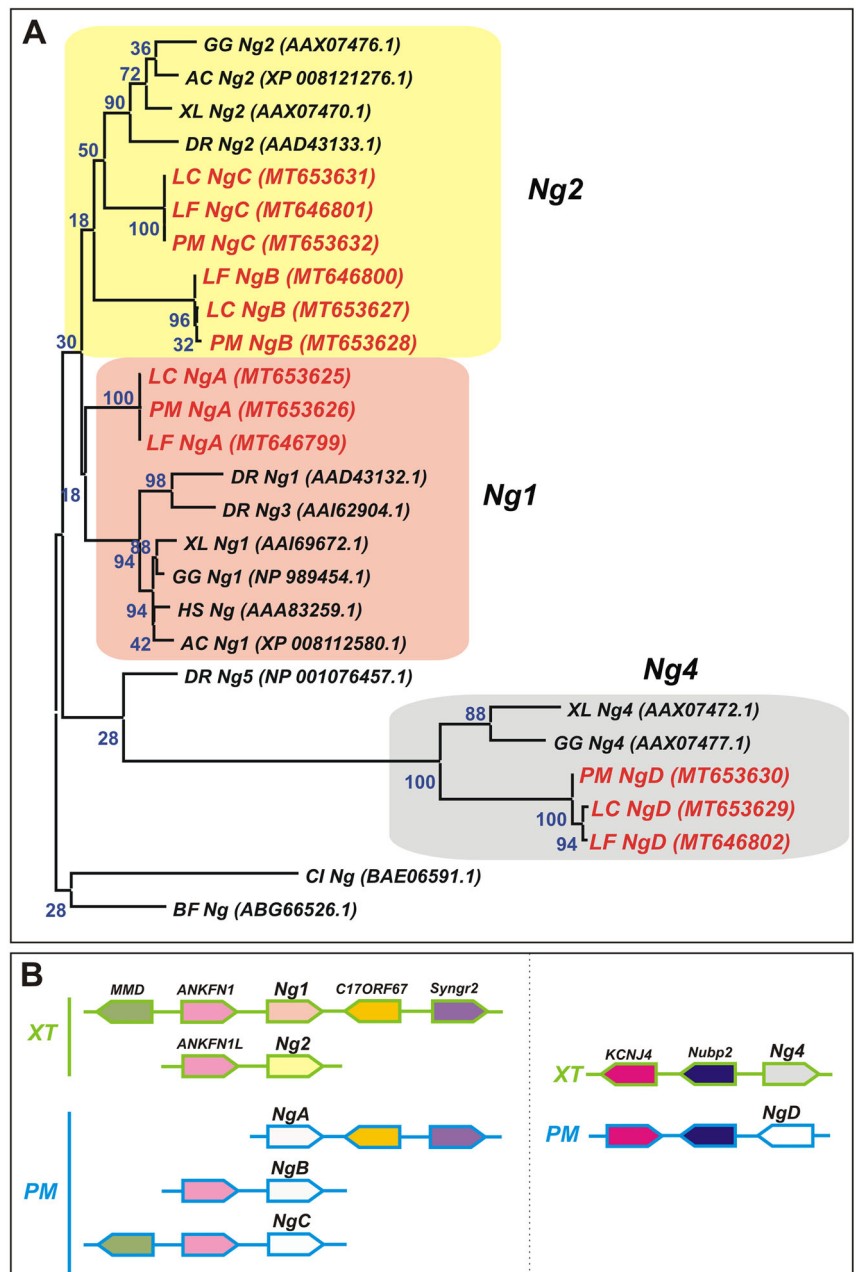

**Fig. 1 Phylogenetic analysis (a) and synteny (b) of lamprey *Noggins*. A** - Phylogenetic tree of *Noggins* is constructed using the Maximum Likehood algorithm. AC *Anolis carolinensis*, BF *Branchiostoma floridae*, CI *Ciona intestinalis*, DR *Danio rerio*, GG *Gallus gallus*, HS *Homo sapiens*, LF *Lampetra fluviatilis*, LC *Lethenteron camtschaticum*, PM *Petromyzon marinus* XL *Xenopus laevis*, XT *Xenopus tropicalis*. **B** - synteny of *Noggin* genes of sea lamprey *P. marinus* (PM) and western clawed frog *X. tropicalis* (XT).

that flank *X. tropicalis Noggin4*, also flank lamprey *NogginD*. On the basis of the data of the ML protein analysis and the fact that, similar to *Noggin4*, *NogginD* has characteristic amino acid substitutions putatively preventing its binding to BMP, one may confidently conclude that the lamprey *NogginD* is indeed the orthologue of *Noggin4* in gnathostomes.

**The expression of lamprey *Noggins* in early development.** To compare the lamprey *Noggins* with their homologs of gnathostomes and to predict their physiological roles in early development, we investigated their temporal and spatial expression patterns.

The analysis of the temporal patterns of *Noggins'* expression was performed by real-time qPCR at the early stages of European

river lamprey (*L. fluviatilis*) development, starting from stage 9 and continuing to stage 26, according to ref. [23].

As a result, we established that *NogginA*, *NogginB* and *NogginC* are expressed after fertilization at very low levels; however, their expression begins to increase starting from the late neurula stage - stage 20 (Fig. 2). In contrast, *NogginD* is expressed at approximately equal levels at all tested stages (Fig. 2).

The spatial expression patterns of *Noggins* were analysed by whole-mount *in situ* hybridization. In these experiments, we analysed *Noggin* expression beginning from stage 11 (late blastula stage according to ref. [23]) and to the stage of pre-ammocoete larvae (stage 29), using anti-sense DIG-labelled probes of mRNA for each of *Noggin* genes of *L. fluviatilis*. Sense DIG-labelled *Noggin* mRNAs were used as the controls.

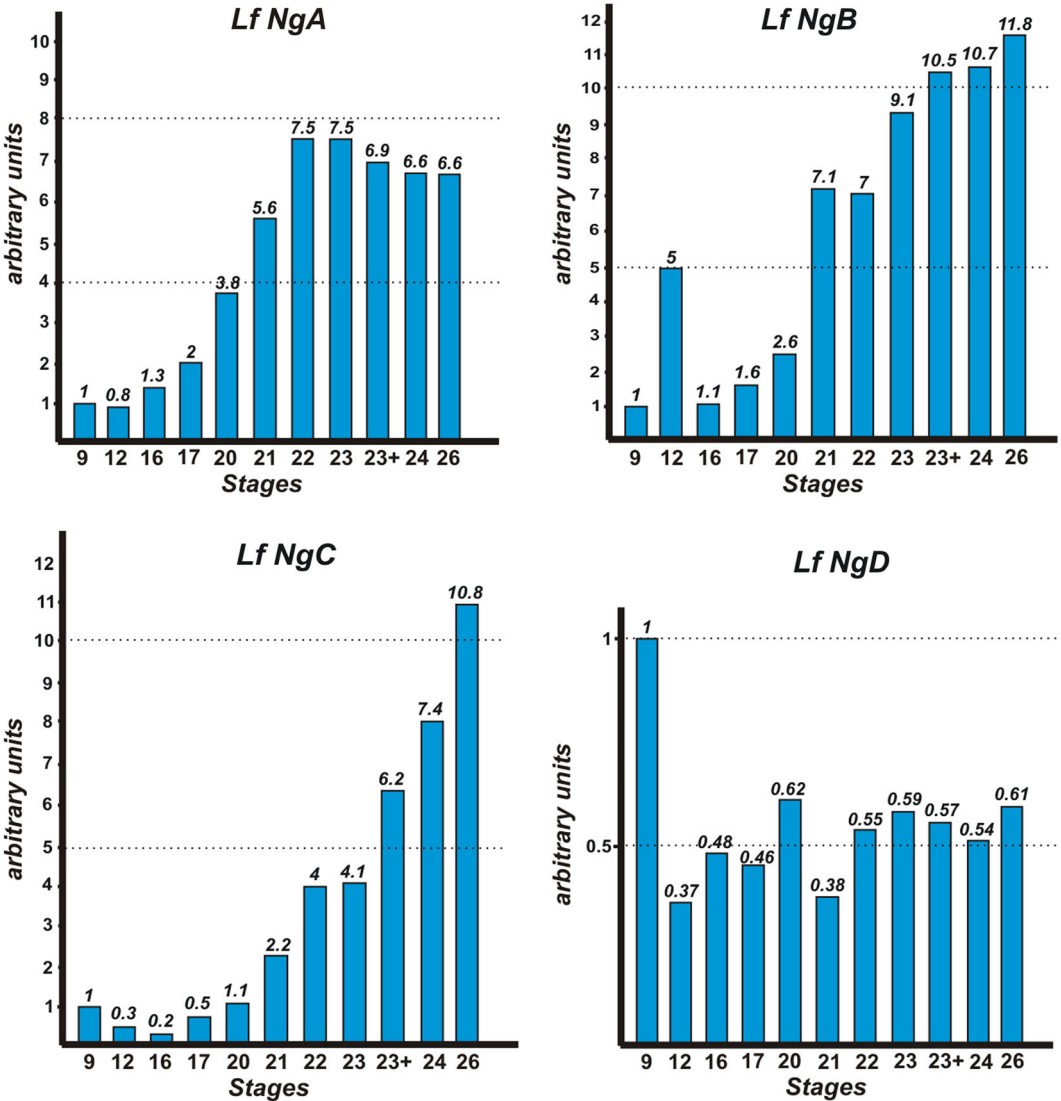

**Fig. 2 The expression dynamics of NogginA, NogginB, NogginC and NogginD in the early stages of development of *L. fluviatilis*.** Stage numbers are indicated according to ref. [24].

Notably, none of the lamprey *Noggin* genes were expressed at the blastula or gastrula stages. In later stages, the analysis yielded the following results.

*NogginA* (Fig. 3a–j, Fig. 4a–j).

*NogginA* expression was detected for the first time at the early neurula stage (st. 17) in the caudal chordamesoderm (Fig. 3a). At the late neurula stage (st. 19–20), the expression of this gene was found in the entire chordamesoderm, except for its most anterior region (Fig. 3b, c). In addition, *NogginA* expression was observed in the dorsal region of the presumptive brain, excluding the most anterior part (Fig. 3b).

At the stage of head outgrowth (st. 22), the expression is clearly visible in the floor plate of the neural tube, in the notochord, in the somites, in the cheek process and in the diencephalon, midbrain and hindbrain (Fig. 3d).

The expression pattern stabilizes at stages 23–25 and includes somites, upper and lower lips, notochord (except for the anterior part) and hypochord (Figs. 3e–h, 4b, c). As observed with transverse sections, at these stages, the expression is localized at the dorsal and ventral edges of the somites. In the spinal cord, the cells of the ventricular zone are stained (Fig. 4a–c). In the brain,

the expression of *NogginA* is found in several areas: at the border of telencephalon and diencephalon, in the region of *zona limitans intrathalamica* (ZLI) that separates the ventral and dorsal parts of the diencephalon and at the border of the midbrain and hindbrain (Figs. 3i, 4d, e). Importantly, since ZLI and the mid-hind brain boundary are known to be secondary organizers of the developing brain, one may suppose that *NogginA* plays a role as one of the factors secreted by these organizers. In addition, the expression of *NogginA* is seen in the borders of some rhombomeres in the hindbrain. In all these brain areas, expression is found in cells of the roof and the floor plates of the neural tube. At stage 27, the expression appears in the developing eyes and in the pharyngeal arcs (Fig. 4f, g).

Interestingly, in the course of development, the pattern of *NogginA* expression in the notochord changes in the following way. At the early stages (st. 17–21), this gene is expressed throughout the notochord. Then, from stage 22, the expression disappears in the anterior part of the notochord while remaining strong in the region of the hindbrain and gradually weakening towards the caudal part of the embryo. Finally, starting at the tail bud stage (st. 26–27), an increase of the expression in the

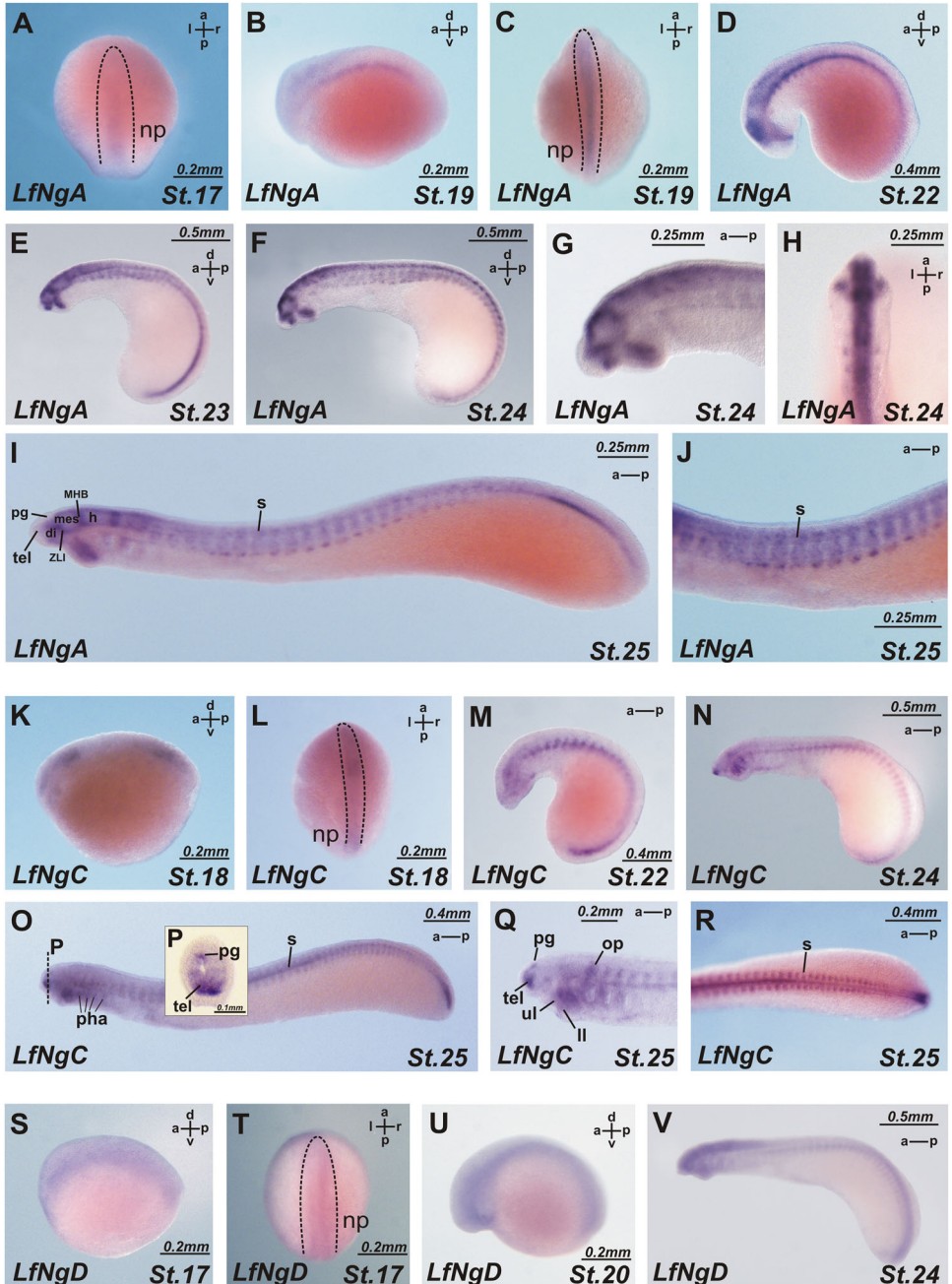

**Fig. 3 Expression patterns of *L. fluviatilis NogginA* (a–j), *NogginC* (k–r) and *NogginD* (s–v). a** At stage 17 *NogginA* is expressed in the caudal chordamesoderm. **b, c** At stage 19 expression is in the entire chordamesoderm except for its most anterior part and in the dorsal region of the presumptive brain, excluding the most anterior part. **d** At stage 22 expression is in the floor and roof plate of the neural tube, in the notochord, in the somites, in the cheek process and in the diencephalon, midbrain and hindbrain. **e–j** At stages 23–25 expression pattern includes somites (**i, j**), upper and lower lips (**g**), notochord, in several areas of the brain: at the border of telencephalon and diencephalon, in *zona limitans intrathalamica*, at mid-hind brain boundary and in the hindbrain (**g, i**). **k, l** At stage 18 *NogginC* is expressed in the 2 chordamesoderm regions: anterior part and caudal part. **m** At stage 22 *NogginC* is found in the notochord, neural tube, pharyngeal arches, otic pits, cheek process and somites. **n** At stages 24 the expression of *NogginC* appears in the forebrain and otic visicles, upper and lower lips. **o–q** At stages 25 the cells of the pineal gland (transverse section on P) also express *NogginC*. **r** *NogginC* expression in somites at stage 25. **s, t** At stage 17 *NogginD* is expressed in the neural tube. **u, v** At stage 17 the expression of *NogginD* develops throughout the neural system in a diffuse manner and weakly in somites. as anterior intraencephalic sulcus, ch optic chiasma; di diencephalon, e eye, fb forebrain, h hindbrain, hb habenula, hpt hypothalamus, ll lower lip, ul upper lip, mes mesencephalon, MHB mid-hindbrain boundary, n notochord, ncc neural crest cells, np neural plate, nt neural tube, oc otic cup, op olfactory placode, pg pineal gland, pha pharyngeal arche, tel telencephalon, ZLI *zona limitans intrathalamica*.

notochord of the growing tail is observed (Fig. 4d, h–j). The latter expression pattern is maintained until stage 29.

*NogginB* (Fig. 5).

The initial expression of *NogginB* was revealed by *in situ* hybridization at the early neurula stage (st. 17) in the anterior end

of the neural tube (Fig. 5a, e). Presumably, these cells will develop into the neural crest or telencephalon. At the mid neurula stage (st. 18), the expression is enhanced in the entire presumptive forebrain and hindbrain and remains in the ventral part of the presumptive midbrain (Fig. 5b, f). By the late neurula stage (st.

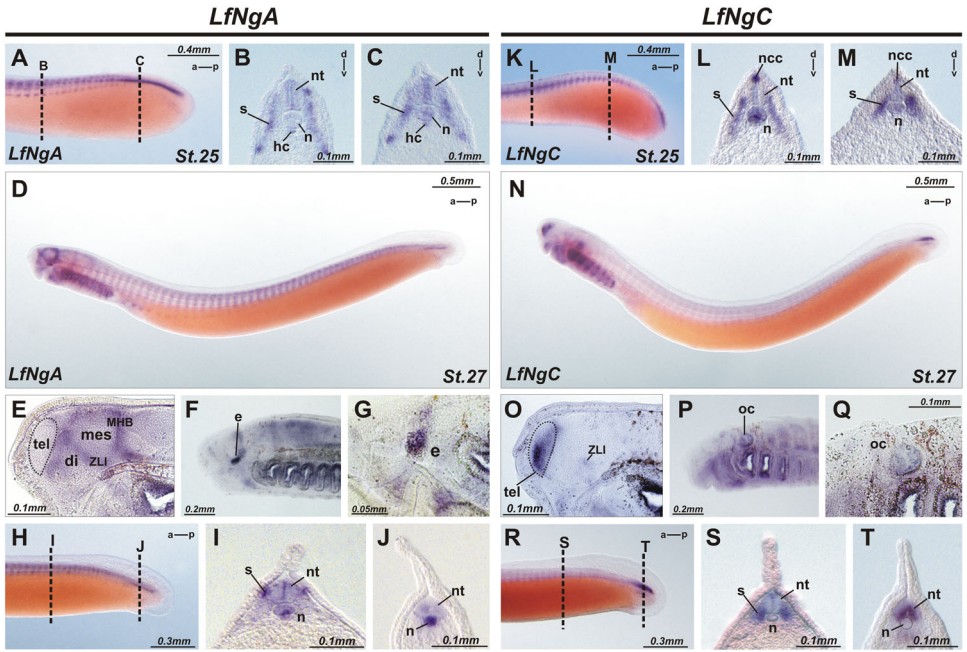

**Fig. 4 Matching of expression patterns of *NogginA* (a–j) and *NogginC* (k–t). a** At stage 25 *NogginA* is expressed in somites, especially in dorsal and ventral angles, notochord, hypochord, neural tube. **b**, **c** Transverse sections, the section levels are shown in (**a**). **d** Whole mount expression of *NogginA* at stage 27. **e**–**g** Saggital section of embryo of stage 27 reveals the expression of *NogginA* at the border of telencephalon and diencephalon along anterior intraencephalic sulcus, in *zona limitans intrathalamica* and at mid-hind brain boundary and in the eyes. **h**–**j** In the tailbud the expression of *NogginA* increases in the notochord of the growing tail. The levels of transverse sections are shown in (**h**). **k**–**m** At stage 25 *NogginC* is expressed in somites, notochord, neural tube and premigratory neural crest cells. **l**, **m** Transverse sections, the section levels are shown in (**k**). **n** Whole mount expression of *NogginC* at stage27. **o**–**q** Saggital sections reveal the stronger expression in the ventral part of the telencephalon and weak expression in ZLI and in otic cups. **r**–**t** at stage 27 *NogginC* is expressed in somites, notochord, neural tube at the trunk level (**s**), but in the growing tail (**t**) it reveals at a high level in the central (ventricular) zone of the neural tube. The levels of transverse sections are shown in (**r**). For abbreviations see Fig. 3.

19–20), the expression of *NogginB* increases in the forebrain, while its expression in the dorsal part of the hindbrain spreads to the posterior and weakens (Fig. 5c, g).

At the head outgrowth stage (st. 21–23), *NogginB* is expressed in the region of the presumptive forebrain and in individual cells located on the dorsal side of the neural tube from the head to the tail, presumably - premigratory neural crest cells (Fig. 5d, h, n). At stages 24–29, strong expression persists in the dorsal and ventral parts of the telencephalon and in the preoptic region of the hypothalamus (Fig. 5s–w).

In the neural crest cells on the dorsal side of the neural tube, *NogginB* is expressed to stage 27 symmetrically in individual cells in the mid-hindbrain and in the spinal cord (Fig. 5o–y)

In summary, one may say that *NogginB* is predominantly expressed in the evolutionarily younger structures, namely, in cells of the neural crest and the telencephalon, which during evolution, appeared for the first time in vertebrates.

*NogginC* (Figs. 3k–r, 4k–t)

At the neurula stage (st. 18–19), the expression of *NogginC* appears in two chordamesoderm domains: one in the future head and one in the caudal region (Fig. 3k, l). Subsequently, in the late neurula stage (st. 20), the expression is detected in the chordomesoderm of the trunk, in the hindbrain region and in the cells of the neural crest.

At the head outgrowth stage (st. 22), *NogginC* is found in the trunk part of the notochord, especially in the cells of its tail region, in the somites, pharyngeal arches, cheek process, otic pits, and cells of the neural crest (Figs. 3m, 4l, m). Later, at stages 24–25, the expression of *NogginC* appears in the forebrain and in the pineal gland (Fig. 3n–r). Weak staining was also observed in the ZLI region of the diencephalon.

At the tailbud stage (st. 27), the expression of *NogginC* in the telencephalon becomes more pronounced in the ventral part and gradually decreases towards the dorsal part (Fig. 4o). Expression in the ears was detected, but the endolymphatic duct was not stained (Fig. 4p, q). In the growing tail, a high level of expression was observed in the central (ventricular) zone of the neural tube, where neural cells are known to actively proliferate (Fig. 4r–t). This pattern of expression is maintained until stage 29.

Thus, it is notable that the expression pattern of *NogginC* partially overlaps the expression patterns of *NogginA* and *NogginB*. The expression in unique vertebrate structures— telencephalon and neural crest cells—was similar for *NogginC* and *NogginB*.

*NogginD* (Fig. 3s–v).

The expression of *NogginD*, similar to the expression of other lamprey *Noggin* genes, is first detected starting from the early neurula stage (st. 17), but in contrast to the latter, it has a diffuse character, making it uniformly distributed throughout the neural plate (Fig. 3s, t). At the next stages of development, up to the pre-ammocoete stage (st. 29), we also observed weak diffuse staining throughout the neural system and somites (Fig. 3u, v). No expression was found in the notochord.

In general, it can be noted that the expression patterns of the lamprey *Noggin* genes have many similarities with the expression patterns of their homologues in other vertebrates, particularly those of *X. laevis*[17]. Thus, the expression of *NogginA* in the diencephalon, midbrain and hindbrain, as well as in the mesoderm derivatives, notochord and somites, appears to be similar to the expression of *Xenopus Noggin1*. The expression of *NogginB* in the forebrain is similar to that of *Noggin2*. At the same time, the expression pattern of *NogginC* has many overlapping

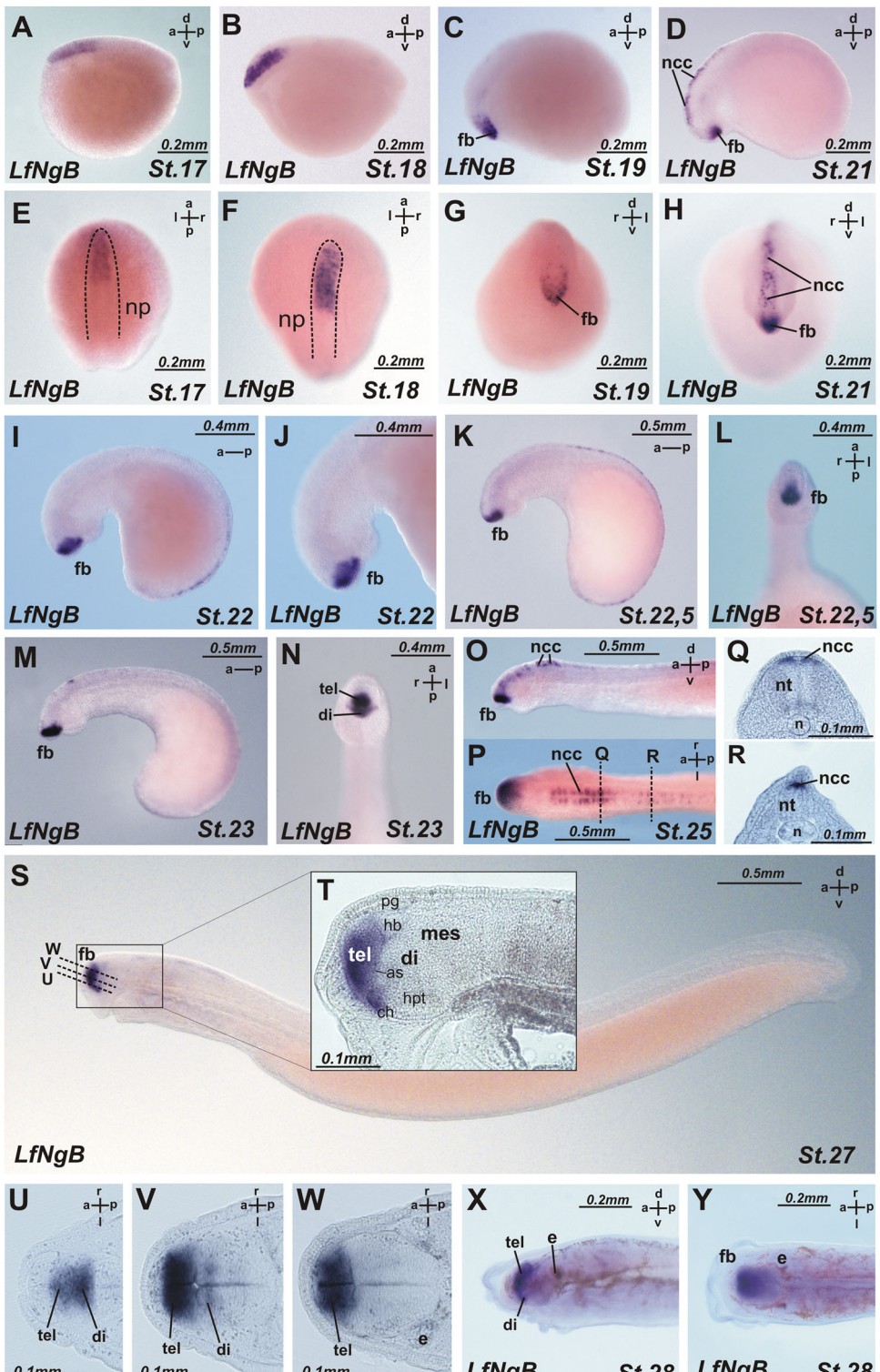

**Fig. 5 Expression pattern of *L. fluviatilis NogginB*. a, e** At stage 17 *NogginB* is expressed in the anterior end of the neural tube. **b, f** At stage 18 *its expression* is enhanced in the presumptive forebrain and hindbrain and weakens in the dorsal part of the presumptive midbrain. **c, d, g, h** At stage 19–21 the *NogginB* expression focuses on the presumptive forebrain (**c, g**) and in the most dorsal cells of neural tube (D, H) presumably - premigratory neural crest cells. **i–r** At stage 22–25 *NogginB* expression persists in the telencephalon and in ventral diencephalon and as is shown on transverse sections (**q, r**) in the premigratory neural crest cells. **s** Whole mount expression of *NogginB* at stage 27. **t** Saggital section of embyo of stage 27 reveal *NogginB* expression in the dorsal and ventral parts of the telencephalon and in the preoptic region of the hypothalamus. **u–w** horizontal section of embyo of stage 27. The section levels are shown in **S**. **x, y** *NogginB* expression at stage 28. For abbreviations see Fig. 3.

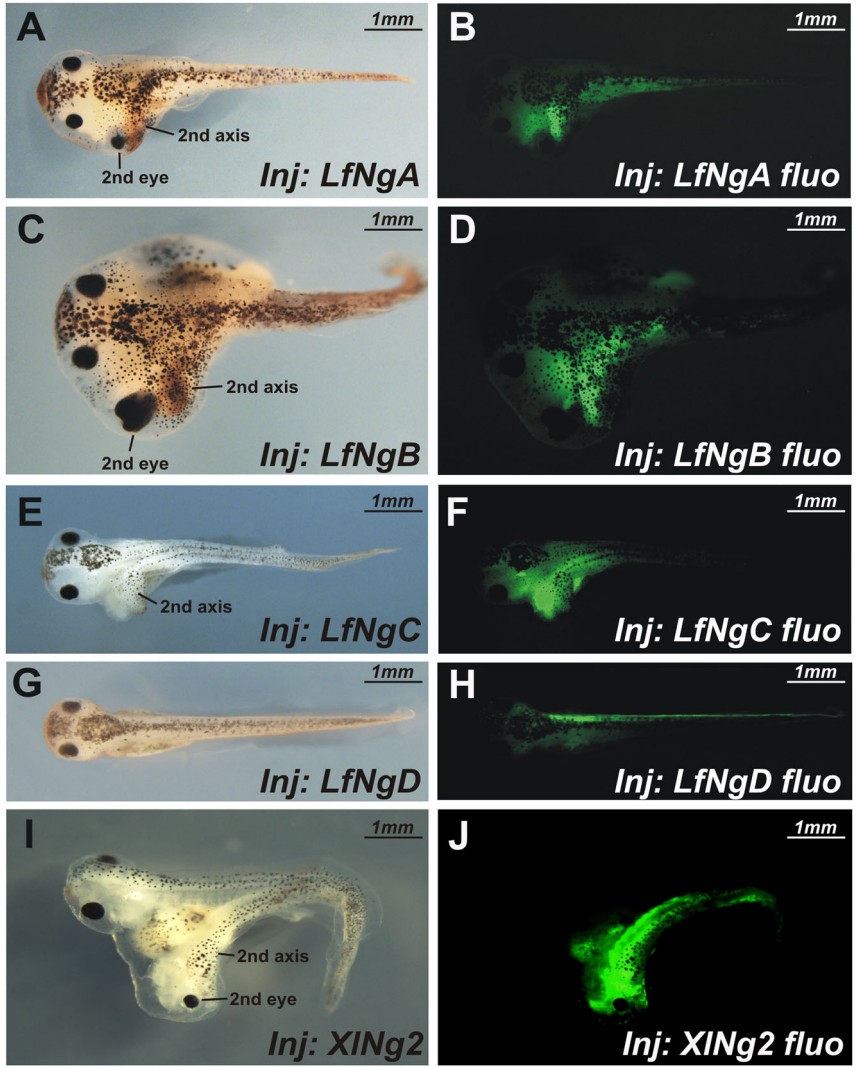

**Fig. 6 Functional conservatism of *Noggins* in vertebrates.** Microinjections of mRNAs of lamprey *NogginA* (**a**, **b**), *NogginB* (**c**, **d**) and *NogginC* (**e**, **f**) in embryos of *X. laevis* causes the formation of secondary axes including anterior-headed structures like mRNA of *X. laevis Noggin2* (**i**, **j**). For percentage of secondary axis and secondary head structures see Supplementary Figure 3. *NogginD* (**g**, **h**) does not induce secondary axes.

features with the expression patterns of *NogginA* and *NogginB*. Finally, the expression of *NogginD*, with its characteristic diffuse pattern, greatly resembles that of *Noggin4*.

**Lamprey Noggins induce secondary body axes in *X. laevis* embryos.** To understand the extent to which the physiological functions of the lamprey Noggins are similar to those of previously described Noggins in other vertebrates, we examined their ability to induce the formation of an additional body axis in *X. laevis* embryos injected ventrally at the 8-cell blastomere stage with mRNAs encoding for these four lamprey's Noggins. The ability to form additional axial structures in the case of ventral expression in the *X. laevis* embryos was shown previously for *Xenopus* Noggin1 and Noggin2[15,24,25]. In contrast, Noggin4 does not have this ability due to its inability to bind BMP[16].

As a result, we found that NogginA, NogginB, and NogginC caused the formation of additional body axes, which in many cases, led to a forebrain with eyes (Fig. 6, Supplementary Fig. 2). Interestingly, while NogginB induced the development of a forebrain with eyes in the highest percentage of cases (13%, $n = 92$), NogginC generated secondary axes with a forebrain and eyes in the lowest percentage of cases (2%, $n = 98$), while NogginA

occupied the middle position according to this criterion (5%, $n = 87$). These findings confirm the ability of the lamprey Noggins to induce secondary axes and are consistent with the observed homology of these proteins with the vertebrate Noggin1 and Noggin2, which have similar properties, as described previously[15].

At the same time, NogginD, which is the presumptive orthologue of Noggin4, was unable to induce secondary axes, similar to the latter[16] (Fig. 6g, h).

## Discussion

**Orthology of lamprey and gnathostomes *Noggins*.** In this work, we described four *Noggin* family genes in lampreys, named *NogginA*, *NogginB*, *NogginC* and *NogginD*. As was revealed by phylogenetic analysis, NogginA gravitates to gnathostome Noggin1, NogginB and NogginC to Noggin2, and NogginD to Noggin4 (Fig. 1a). Despite the fact that the bootstrap value is rather low (50% or even less), such phylogenetic relationships of these three proteins look quite consistent with the local genomic synteny of their genes. Namely, *NogginB*, *NogginC* and *Noggin2* have *ANKFN1* adjoined at the 5' end, while neither *NogginA* nor *NogginD* have this gene anywhere in their respective vicinity.

At the same time, the analysis of genomic synteny revealed a relationship of lamprey *NogginA*, *NogginB* *NogginC*, and gnathostome *Noggin2*, with gnathostome *Noggin1*, which is flanked by the homologue of the 5'-end neighbour of *NogginB*, *NogginC* and *Noggin2* (*ANKFN1*) and the homologue of the 3'-end neighbour of *NogginA* (*C17ORF67* and *Syngr2*) (Fig. 1b). Obviously, this finding indicates that all five genes could have been derived from a common ancestral *Noggin1*-like gene of vertebrates, which existed before the jawless and gnathostomes separated. In addition, the origin of *NogginA*, *NogginB*, *NogginC*, *Noggin1* and *Noggin2* from a common ancestor is confirmed by the following independent data. According to genome sequencing, *P. marinus NogginA*, *NogginB*, and *NogginC* are located in superscaffold 00003, 00037 and 00011, respectively. On the other hand, an analysis of the distribution of conserved syntenies in lamprey and chicken genomes, conducted by Smith et al., indicated that these superscaffolds correspond to chicken chromosome 14 and 18, which the authors believe likely emerged due to the duplication of one ancestral chromosome[5]. Meanwhile, chicken *Noggin1* and *Noggin2* genes are precisely located on chromosome 18 and 14, respectively. Thus, in good agreement with our results, these data suggest that at least three of four lamprey *Noggin* genes (i.e., *NogginA*, *NogginB*, and *NogginC*), along with gnathostome *Noggin1* and *Noggin2*, may indeed have been derived from a single ancestral gene.

At the same time, the synteny of *NogginD* with *Noggin4* of gnathostomes, together with the lack of their synteny with any other vertebrate *Noggin* gene, suggests that either these genes were derived from some other *Noggin* gene, which existed before the emergence of the vertebrate branch, or that the ancestor of *NogginD* and *Noggin4* appeared as a result of a local translocation of a copy of a *Noggin1*- or *Noggin2*-type gene into different genomic locations. However, since all deuterostome invertebrates with sequenced genomes have only one *Noggin*, most resembling *Noggin1* and *Noggin2*, but lack *Noggin4*, the second scenario, which suggests the appearance of a *NogginD/Noggin4* ancestor during the initial period of vertebrate evolution, before the splitting of the jawless fish and gnathostomes, seems to us more realistic (see below).

**Noggins have conserved expression patterns in vertebrates**. Lamprey embryos are very attractive models for evo-devo studies because, among that of all extant vertebrates, the lamprey evolutionary branch was separated from the common trunk at the earliest stages of vertebrate evolution. Hence, it is presumed that the lamprey genes could have preserved the types of expression patterns characteristic of ancestral vertebrates.

Previous studies of the *Noggin* family genes in vertebrates have shown that all three *Noggins* identified in this group (*Noggin1*, *Noggin2* and *Noggin4*) differ in their expression patterns and functional properties.

The expression of the "classic" *Noggin1*, as well as of that of *Noggin2* and *Noggin4*, was studied in detail in *Xenopus*[9,17,18]. In this species, the expression of *Noggin1* is detected for the first time at the beginning of gastrulation in the area of the dorsal blastopore lip, i.e., in the Spemann organizer, and then continues in the notochord, the midline of the neural plate and in the stripe of cells surrounding the anterior margin of the neural plate[9]. At later stages, this gene is expressed in the notochord, spinal cord, optic vesicle, branchial, pharyngeal and mandibular arches. Weaker expression is detected in the forebrain and in the head mesenchyme[9].

In contrast to that of *Noggin1*, the expression of *Noggin2* starts in *X. laevis* only from the early neurula stage, in the stripe of cells at the anterior margin of the neural plate; thus, it is partially

superimposed with the expression of *Noggin1*. In contrast to *Noggin1*, however, no expression of *Noggin2* was detected in the notochord[17,18]. At later stages, the expression at the highest level was observed in the dorsal region of the developing forebrain and at a lower level in the dorsal parts of the hindbrain, dorsal parts of the somites and in the forming heart. In addition, the expression of *Noggin2* was detected in the derivatives of the neural crest, the pharyngeal arches[17,18].

A specific feature of the expression of *Noggin4* is its very diffuse distribution in embryonic tissues. Namely, in *X. laevis*, it is detected for the first time at the gastrula stage in the epidermal layer of the entire animal hemisphere[18]. At the neurula stage, *Noggin4* expression is observed throughout the neural plate, with the maximum level in its anterior part, in the presumptive area of the head placodes and in the region of the future neural crest. In later stages, *Noggin4* is expressed in the epidermis covering the neural tube, including the forebrain and spinal cord, in the auditory vesicles, cement gland and neural crest derivatives, including gill arches. Thin stripes of expression were also observed along the borders of the somites[18].

In general, such expression patterns of *Xenopus Noggin* genes demonstrate many common features with the expression patterns of their lamprey orthologues.

Thus, lamprey *NogginA*, which begins to be expressed in lamprey somewhat later than *Noggin1* in *Xenopus*, is also expressed in the chordamesoderm and later in the somites and head structures, including the neural crest derivatives.

*NogginB*, similar to its *Xenopus* orthologue, *Noggin2*, is also first detected at the early neurula stage in the anterior part of the neural tube and is subsequently expressed in the telencephalon and neural crest cells but not in the notochord.

*NogginC* has expression features in common both with those of both *NogginA* (expression in mesodermal derivatives) and *NogginB* (expression in telencephalon and neural crest cells).

*NogginD*, similar to *Noggin4*, has a diffuse expression pattern observed from the beginning of the neurula stage in the neural plate and later throughout the neural system.

The similarities are also revealed at the functional level. The ectopic expression of *NogginA*, *NogginB* and *NogginC* in *X. laevis* embryos induces the development of additional axes containing the forebrain with eyes, similar to those induced by ectopic expression of their *Xenopus* counterparts. In contrast, *NogginD*, like its *Xenopus* orthologue *Noggin4*, does not demonstrate this ability. Such conservatism indicates the involvement of *Noggin* genes in both jawless fish and gnathostomes in mechanisms regulating the early development of mainly the same anatomical structures. In turn, this observation also suggests that these mechanisms may have appeared before the divergence of the jawless and gnathostomes; thus, they are likely part of the basic regulatory network that was formed in the vertebrate ancestors and became critical for the formation of the body plan traits specific for all vertebrates. Notably, the same conclusions were previously made for some other regulatory genes after a comparative analysis of their expression patterns in lampreys and gnathostomes[5,26–28].

In this respect, the expression of lamprey *NogginB* and *Xenopus Noggin2* in such a unique vertebrate structure as the rudimentary telencephalon looks extremely exciting. Indeed, given the results of our phylogenetic analysis that indicate the origin of these genes for the first time only in vertebrates to our knowledge, one may suppose that their emergence at the beginning of vertebrate evolution could be one of the necessary prerequisites for the appearance of the telencephalon in vertebrates. Importantly, a similar role was proposed earlier for the homeobox gene *Anf/Hesx1*, which also emerged for the first time only in vertebrates and the expression of which is indicated

during early development of all vertebrates, including lampreys, in the rudimentary telencephalon[29–32]. Our present data showing that the expression of another telencephalon-specific gene, *NogginB/Noggin2*, which also appeared for the first time in the ancestor of vertebrates, confirms that the emergence of novel genes could have indeed played an important role in acquiring unique anatomical innovations in this phylum of animals.

**Hypothesis of *Noggins* evolution in vertebrates.** As is currently discussed in the scientific community, this ancient period of vertebrate evolution included either one or two rounds of ancestral whole-genome duplication[8,33,34]. Initially, this supposition was inspired by the finding of four and six complexes of *HOX* genes in gnathostomes and lampreys, respectively[35–37]. At the same time, the nearest relatives of vertebrates, tunicates and cephalochordates, have only one *HOX* complex. Thereafter, similar results were obtained for the *Pax6* gene[38]. These findings indicated that either two rounds of genome duplication occurred before the divergence of the jawless fish and gnathostomes, followed by an additional round of the entire or partial genomic duplications in the jawless fish, or that only one round of the duplication preceded the divergence of the jawless fish and gnathostomes, with additional rounds occurring independently in these two branches of vertebrates[7,8,39].

The revealed phylogeny, synteny, expression patterns and functional analyses show that vertebrates *Noggins* can be confidently divided into two clusters: *NogginA/B/C/1/2* and *NogginD/4*. At the same time, the expression pattern of lamprey *NogginA* has many similarities with jawed *Noggin1*, while *NogginB* has the expression pattern very similar to *Noggin2*. As the independent origin of so similar patterns in different lineages looks in our opinion very unlikely, one may hypothesize that the common vertebrate ancestor should have had at least three *Noggin* genes before the split of cyclostomes and gnathostomes. Based on these considerations, our data are not consistent with the recently proposed model suggesting only one common duplication before the split of Cyclostomes and Gnathostomes followed by additional independent duplications after separation of these lineages[40]. It seems more logical to suppose that the appearance of at least three common functionally diverged *Noggins* in the cyclostome and gnathostome common ancestor is consistent with two general scenarios: (a) one "basal" *Noggin* gene in vertebrate ancestor passed through two rounds of duplication before the divergence of the jawless fishes and gnathostomes; (b) two "basal" genes (obviously *NogginA/B/C/1/2* and *NogginD/4*) passed through one round of duplication in the common ancestor. Let us consider both these models.

If two rounds of whole-genome duplications preceded the divergence of cyclostomes and gnathostomes, one may hypothesize that, during the first round, two copies of ancestral *Noggin* arose, followed by the evolution of one of these copies into ancestral *NogginB/C/2* (Fig. 7a). The latter gene might have inherited at least two 5′ genomic neighbours of the ancestral *Noggin1* (*MMD* and *ANKFN1*), which are still present in the vicinity of *NogginC*. Then, two copies of ancestral *NogginA/1* and *NogginB/C/2* could have appeared as a result of the second round of whole-genome duplication. One copy of ancestral *NogginB/C/2* could have lost the homologue of *MMD*, thus acquiring the specific 3′ region surrounding the evolved *NogginB* and *Noggin2* genes (the ancestor of *NogginB/2*). At the same time, the second copy of ancestral *NogginB/C/2*, which maintained both *MMD* and *ANKFN1* near the 5′ end, became the ancestor of *NogginC*. Thereafter, one of two copies of ancestral *NogginA/1* might have been translocated to a different genomic location, followed by specific point mutations in this translocated copy, which impaired

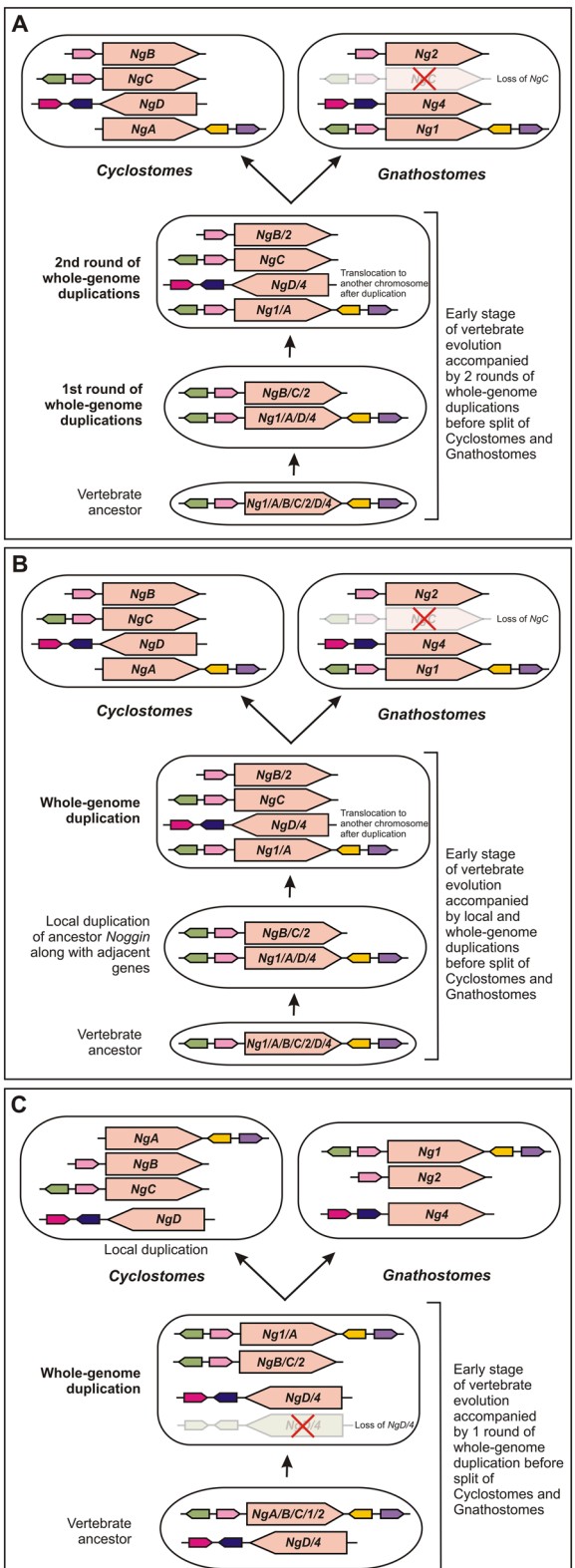

**Fig. 7 Three possible scenarios of the evolution of *Noggin* genes in vertebrates. a** and **b** scenarios suggest one ancestral *Noggin* gene passed through two rounds of duplication (2 rounds of whole-genome duplication in **a** vs local+whole-genome duplications in **b**). **c** scenario suggests two ancestral *Noggins* passed through one round of whole-genome duplication.

the ability of its protein product to bind BMP. Finally, further mutagenesis could have been generated the ancestor of NogginD/4 from this translocated copy. After the subsequent separation of the jawless and gnathostomes, extant lampreys inherited all four Noggin genes: A, B, C and D. At the same time, in the vertebrate branch, the ancestor of NogginC seemingly was lost, resulting in only three Noggin genes now present in most of the extant vertebrates: Noggin1, Noggin2 and Noggin4.

A possible variant of this scenario could include local duplication of the ancestor Noggin gene along with its adjacent genes instead of a whole-genome duplication at the first step (Fig. 7b). Based on the data of the present work, it is not possible to judge whether the first round of duplication was whole-genome or local.

In case of the one-round scenario of whole-genome duplication, one has to allow the presence of two Noggin genes in vertebrate ancestor (Fig. 7c). After the single round of genomic duplication, vertebrates could obtain NogginA/1, NogginB/C/2 and two copies of Noggin4/D genes. As all extant vertebrates have only one copy of NogginD or Noggin4 gene, another copy of this ancestor Noggin4/D had to disappear somewhere in between the moment of the whole-genome duplication and the moment of the divergence of jawless and gnathostomes. The appearance of NogginB and NogginC in this model could be the result of a local duplication and further divergence of the descendants of NogginB/C/2 in lampreys.

In support of the hypothesis of the existence of two "basic" Noggins in vertebrate ancestor one may point to the presence of multiple noggin-like genes (nlg) in the flat worm Schmidtea mediterrania[41,42], which could be considered as possible predecessors of the hypothetical NogginA/B/C/1/2 and NogginD/4 in the vertebrate ancestor. As was shown by the authors using X. laevis embryos as models, at least some of the nlgs were indeed able to bind BMP, like vertebrate Noggin1 and 2, while other not, like Noggin4. However, on our opinion, it is very difficult to imagine direct phylogenetic relationship of the flat worm nlgs with vertebrate Noggin4 because of the aforementioned lack of Noggin4 homologs genes in all extant deuterostome predecessors of vertebrates that separate them from flat worms. The independent appearance of Noggin4 and noggin-like genes whose protein products lack the ability to bind BMP could be based on the emergence of multiple copies of Noggin genes in any animal lineage. In this case, one copy of Noggin may continue to retain the ability to bind BMP, while another copy(s) could escape from the pressure of stabilizing natural selection and quickly acquire point mutations in any one of the four amino acid positions, which have been shown to be critical for BMP binding[14]. Such mutant copy(s) of Noggin may not have been eliminated during evolution because, as we previously showed, Noggin proteins, in addition to BMP binding, can bind ligands of other signalling pathways, such as those of the Wnt and Activin/Nodal pathways, thereby participating in the regulation of other important developmental processes[15,16].

Given all these considerations and being guided by Occam's razor principle, we believe that at the present time the hypothesis of two rounds of genome duplications before the divergence of the jawless fish and gnathostomes may explain less controversially the presence of those Noggins that were revealed in lampreys and gnathostomes.

## Methods

**Animals**. All animal experiments were performed in accordance with guidelines approved by the Shemyakin-Ovchinnikov Institute of Bioorganic Chemistry (Moscow, Russia) Animal Committee and handled in accordance with the 1986 Animals (Scientific Procedures) Act and Helsinki Declaration.

L. fluviatilis adult lampreys were collected in the Saint Petersburg district. Embryos were obtained via artificial fertilization of eggs squeezed from pregnant

females. The embryos were staged as described in Tahara, 1988. For in situ hybridization, embryos were fixed in MEMFA (3,7% formaldehyde, 100 mM MOPS, 2 mM EGTA, 1 mM MgSO$_4$), dehydrated in methanol and kept at −20C.

**Cloning of lamprey Noggins cDNAs**. For RT PCR we used samples of total RNA from L. fluviatilis embryos at early stages of development (st. 12–26). Total RNA from full embryos was extracted by NucleoSpin RNA XS Kit by Macherey-Nagel.

Full-length cDNAs of L. fluviatilis Noggin genes were obtained by nested PCR. For primers see Supplementary Information.

The obtained cDNA was cloned in pAL2-T vector, provided by Evrogen (www.evrogen.ru) and cDNA inserts of 3 clones were sequenced. The obtained full-length cDNAs were used for in situ hybridization and for X. laevis embryos injections.

**Bioinformatics and synteny analysis**. Phylogenetic analysis of protein sequences were performed via the maximum likehood[43] methods using the MEGA6[44] program. The percentage of replicate trees in which the associated taxa clustered together in the bootstrap test (1000 replicates) is shown next to the branches[45]. Evolutionary distances were computed using the JTT matrix-based method[46].

For synteny analysis we compared genomic scaffolds containing lamprey and amphibian Noggins by using Petromyzon marinus genome browser (https://simrbase.stowers.org/organism/Petromyzon/marinus), and Xenopus tropicals genome browser (http://www.xenbase.org/common/displayJBrowse.do?data=data/xt9_1).

**RT-PCR**. For qRT-PCR, three groups of the L. fluviatilis embryos were collected obtaining 50 embryos, respectively, from each of the desired stages. Total RNA was extracted using an RNA isolation kit (MASHEREY-NAGEL) according to the manufacturer's protocol. The concentration of the extracted RNA was measured with a Qubit® fluorometer (Invitrogen), while RNA integrity was checked visually via gel electrophoresis.

First-strand preparation, qPCR parameters and primers used are described in Supplementary Information.

Two independent pairs of primers were used for each of Noggin genes to exclude unspecific signals. For the arbitrary unit in Fig. 2 we take the expression level at the earliest investigated stage - the early blastula.

**Synthetic mRNA and in situ hybridization**. Synthetic mRNAs of the lamprey Noggins were prepared with the mMessage Machine SP6 Kit (Ambion).

L. fluviatilis whole-mount in situ hybridization was performed as described in ref. [32].

The lamprey specimens were embedded in 20% gelatin and 35 µm sections were prepared as described in ref. [47] and observed with light microscope Rihert-Jung.

The embryos were cleared with a graded series of glycerol 25, 50, 75% in PBS and 99% at RT and observed with a Leica M205 stereomicroscope.

**Statistics and reproducibility**. All Noggin plasmids insert sequences were verified by Sanger sequencing of three replicates.

Phylogenetic analyses were performed in the MEGA6[44] program. Multiple alignment was performed via ClustalW algorithm with the default parameters.

Each qPCR sample contained total mRNA from 50 L. fluviatilis embryos of the desired stages. Three replicates of sample series were obtained from different animals. The qPCR data were analyzed using the $\Delta\Delta C_t$ method. The geometric mean of expression of two reference housekeeping genes (ODC and EF1alpha) was used for normalization of the target gene expression levels.

The numbers of Xenopus embryos in the functional test is shown in Supplementary Fig. 3.

**Reporting summary**. Further information on research design is available in the Nature Research Reporting Summary linked to this article.

## Data availability

Genome browsers used for synteny analysis: sea lamprey P. marinus genome browser (https://simrbase.stowers.org/organism/Petromyzon/marinus); western clawed frog X. tropicals genome browser (http://www.xenbase.org/common/displayJBrowse.do?data=data/xt9_1)

Lamprey Noggins cDNAs has been deposited in the GenBank database under the following accession numbers:

P. marinus NogginA MT653626, NogginB MT653628, NogginC MT653632, NogginD MT653630;

L. camtchaticum NogginA MT653625, NogginB MT653627, NogginC MT653631, NogginD MT653629;

L. fluviatilis NogginA MT646799, NogginB MT646800, NogginC MT646801, NogginD MT646802.

All the experimental data generated or analyzed during this study are present in the paper. Additional data and research materials related to this paper are available from the corresponding author on reasonable request.

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

## Acknowledgements

This work was supported by RFBR grant 18-04-00015 (A.V.B.). Experiments with qRT-PCR were supported by Russian Scientific Foundation (19-14-00098). Phylogenetic analysis was supported by RFBR grant 18-29-07014 MK (A.G.Z.). *X. laevis* experiments were supported by RFBR grant 20-04-00675A (G.V.E.).

## Author contributions

G.V.E. - embryos collection, in situ hybridization, histology, photography, figures ideas and preparation, writing of the paper; A.V.K. – *L. fluviatilis* animals, embryos; A.G.Z. – *Xenopus* embryos malformations analysis, figures ideas and preparation, writing of the paper, A.V.B. – databases analysis, cDNA cloning, qPCR, phylogenetic analysis, figures ideas and preparation, writing of the paper.

## Competing interests

The authors declare no competing interests.
