## [Peer Review File · Communications Biology]

Reviewers' comments:

Reviewer #1 (Remarks to the Author):

In this manuscript, Ermakova et al. characterise the Noggin genes in lampreys using molecular cloning, phylogenetics and in situ hybridization approaches. By comparison with gnathostomes (jawed vertebrates), they interpret their findings in the context of early vertebrate genome and developmental evolution.

5 main findings are presented:

- i. Sea Lamprey, Arctic lamprey and European river lamprey each have four Noggin genes: designated here as NogginA-D.
- ii. Phylogenetic analysis of protein sequences groups lamprey Noggin genes with those of gnathostomes as follows: NogginA with Noggin1, NogginB and C with Noggin2, and NogginD with Noggin4.
- iii. Patterns of conserved synteny between sea lamprey and *Xenopus* Noggin loci are consistent with the groups that are inferred from the protein phylogenetic analysis.
- iv. Lamprey Noggin genes display complex and dynamic expression domains during embryogenesis, with many similarities to the patterns seen for their gnathostome homologues.
- v. Lamprey Noggin genes are able to induce a secondary body axis when overexpressed in *Xenopus* embryos, as seen for gnathostome Noggin1 and 2 genes. Interestingly, NogginD does not exhibit this capability, similar to gnathostome Noggin 4, which may be related to its sequence divergence in a region critical for BMP binding.

The lampreys represent an important extant group for investigating early vertebrate evolution, due to their phylogenetic position as sister group (along with hagfish) to the jawed vertebrates. Genome assemblies are now available for two lamprey species, but many lamprey developmental genes are yet to be characterized in lamprey. Detailed identification and characterization of such genes in lampreys is important for understanding how developmental gene regulatory programs diversified and became elaborated during vertebrate evolution. Thus, this study is an important contribution to these efforts.

This work has been carried out with a thorough attention to detail, the findings are novel, and the manuscript is well written and presented. Notably, the in situ hybridization and sectioning data are of very high quality, and these are the strongest aspect of the paper. However, I have concerns about the phylogenetic interpretations, which I would like to see addressed before I can fully recommend publication of this work. As detailed below, all I suggest is that the authors modify the strength of some of their claims regarding the phylogenetic correspondence between certain lamprey and gnathostome Noggin genes, as well as adding text to the abstract to highlight their detailed characterization of Noggin expression in lamprey.

Major points:

- i. The phylogenetic tree in Fig. 1A has a high bootstrap value (100) for the grouping of NogginD with Noggin4, which implies orthology between these proteins. This is fine. However, the relationships between the other lamprey Noggin proteins and their gnathostome homologues seem much less clear. For instance, the clade consisting of NogginA and Noggin1 has a low bootstrap value (18). Similarly, the clade consisting of NogginB, NogginC and Noggin2 also has a low bootstrap value (18). Thus, it appears that the relationships between NogginA,B,C and Noggin1,2 are not resolved with any clarity. Furthermore, it is unclear how the local synteny analysis offers any further resolution, besides supporting the grouping of NogginA,B,C with Noggin1,2 and grouping NogginD with Noggin4. Therefore, the claim made in the discussion that 'NogginA corresponds to gnathostome Noggin1, NogginB and NogginC correspond to Noggin2' is not justified and should be toned down. This lack of phylogenetic clarity is not surprising, given the divergence between lamprey and gnathostomes. I do not have a problem with the authors' phylogenetic approach, just with their interpretation of the tree. I suggest the authors modify the strength of their claims regarding direct correspondence between NogginA and Noggin1, and NogginB,C and Noggin2. The discussion of the different possible scenarios leading to the complement of lamprey and gnathostome Noggin genes is speculative but suitably balanced, so does not require alteration.
- ii. The authors do a great job characterizing the expression domains of the lamprey Noggin genes during embryogenesis and comparing these with their gnathostome counterparts, but this is not mentioned in the abstract. These are valuable results and should be referred to in the abstract as a major focus of the paper.

Reviewer #2 (Remarks to the Author):

This manuscript reported four Noggin homolog genes for the first time from lamprey (I don't think so. See below). They conducted phylogenetic tree analysis and genome synteny analysis, qPCR and in situ hybridization on the lamprey embryos. Finally, they did injection experiment by using *Xenopus* embryo whether the lamprey noggin genes can induce secondary body axes. Their main conclusion of this manuscript is that the latest common ancestor of jawless and jawed vertebrates has experienced two round whole genome duplications. This conclusion mainly comes from the gene phylogenetic analysis and gene synteny analysis.

Major comments:

Regarding identification of lamprey Noggin genes and phylogenetic analysis, it has been already published (Kuraku and Kuratani GBE 2011), which is not cited in this manuscript. Kuraku and Kuratani already identified four Noggin genes from sea lamprey genome and conducted phylogenetic tree analysis. In addition, the estimation of the Noggin gene family evolution is different. Especially, Kuraku concluded that Noggin4/D was split BEFORE the 2 round whole genome duplication. The authors should cite this paper and clarify where is the new point of the current study, and where the different conclusion came from.

Gene expression analysis provides new insights on the lamprey Noggin genes. Only minor anatomical corrections are needed (see below).

The injection experiments are clear and quite convincing. However, it is already reported that Noggin genes from more distant invertebrate animals (hydra and planaria) have induction activity for the second axis as they cited (refs. 27, 28). Therefore, it is not so much valuable information from the evolutionary point of view. In addition, this experiment is much less related to the main conclusion.

Minor comments:

L39: The extant cyclostome is NOT the closest to ancestral vertebrates than modern gnathostomes. Because both groups are equally evolved from the latest common ancestors. Extant cyclostomes have many specific traits, which acquired through their lineages.

L126, L149: Please clarify which amino acid substitutions are critical for the BMP binding activity. Also, please provide us whether those substitutions exist or not in the Nodal4/D.

L140, L185, L198 etc.: Please remove the underlines.

L199, L243: intERthalamica -> intRathalamica

Fig.4E, Fig5N: The position of "mes" is wrong. Mesencephalon should be positioned anterior to the MHB.

There are some typos, Russian character and wrong use of italic characters (i.g. L227 "NogginC is expressed"). Please correct those mistakes carefully or use proofreading services.

Reviewer #3 (Remarks to the Author):

The manuscript "Discovery of four Noggin genes in lampreys may suggest two rounds of genome duplication preceding divergence of Agnatha and Gnathostomata" by Ermakova et al makes a compelling case that the noggin gene underwent two rounds of duplication and evolved specific functions before the divergence of vertebrate and lamprey lineages. Other reviewers can likely provide more accurate comments on the details of expression, so I will limit mine to the discussion.

My only major comments are related to necessary modification to the introduction and discussions regarding duplication scenarios. These should be carefully revised to reflect the referenced studies and the data that are available through this study.

1) An important part of the published 1R model is that some/much of the signal that has been traditionally interpreted as supporting a gnathostome 2R is from local duplications that pre-dated (or postdated) a shared whole duplication event, with no subsequent whole genome duplications in the gnathostome lineage. The data presented show limited conserved synteny in the vicinity of noggin, while this supports two rounds of duplication, it cannot differentiate local vs whole genome duplication. Therefore a statement like "This model is in good agreement with the hypothesis suggesting two rounds of whole-genome duplication in the common ancestor of vertebrates before divergence of jawless and gnathostomes." Should be reworded to "This model is in good agreement

with the hypothesis suggesting two rounds of duplication in the common ancestor of vertebrates before divergence of jawless and gnathostomes." OR "This model is in good agreement with the hypothesis suggesting two rounds of local and/or whole genome duplication in the common ancestor of vertebrates before divergence of jawless and gnathostomes."

2) Incorporating this will require careful modification of the discussion section.

3) Regarding this model, it will be important to reiterate that the authors are proposing that the duplications occurred AND the regulatory/functional changes evolved prior to divergence.

4) The findings presented here would seem to be inconsistent with the recently proposed 2R model (PMID: 32313176) that was originally presented as alternative to explain 1R(plus local)-like patterns

Minor comment:

Use of the term "cyclostome" invokes a specific evolutionary model wherein hagfish and lamprey are considered sister taxa, however it should be noted that phylogenetic support for this hypothesis is weak at best (see PMID: 25071211) the authors might reconsider use of the term, especially since they understandably do not include hagfish data presumably due to the fragmentary nature of the available assembly and difficulty in accessing embryos. Cyclostome is a widely used term the definition of which has little bearing on this study, so it may not be absolutely necessary to dwell on that for the purposes of this manuscript.

Sincerely,

Jeremiah Smith

We appreciate reviewers' detailed analysis of our text.

Our detailed responses to Reviewers' comments are below.

Reviewer #1:

In this manuscript, Ermakova et al. characterise the Noggin genes in lampreys using molecular cloning, phylogenetics and in situ hybridization approaches. By comparison with gnathostomes (jawed vertebrates), they interpret their findings in the context of early vertebrate genome and developmental evolution.

5 main findings are presented:

- i. Sea Lamprey, Arctic lamprey and European river lamprey each have four Noggin genes: designated here as NogginA-D.
- ii. Phylogenetic analysis of protein sequences groups lamprey Noggin genes with those of gnathostomes as follows: NogginA with Noggin1, NogginB and C with Noggin2, and NogginD with Noggin4.
- iii. Patterns of conserved synteny between sea lamprey and *Xenopus* Noggin loci are consistent with the groups that are inferred from the protein phylogenetic analysis.
- iv. Lamprey Noggin genes display complex and dynamic expression domains during embryogenesis, with many similarities to the patterns seen for their gnathostome homologues.
- v. Lamprey Noggin genes are able to induce a secondary body axis when overexpressed in *Xenopus* embryos, as seen for gnathostome Noggin1 and 2 genes. Interestingly, NogginD does not exhibit this capability, similar to gnathostome Noggin 4, which may be related to its sequence divergence in a region critical for BMP binding.

The lampreys represent an important extant group for investigating early vertebrate evolution, due to their phylogenetic position as sister group (along with hagfish) to the jawed vertebrates. Genome assemblies are now available for two lamprey species, but many lamprey developmental genes are yet to be characterized in lamprey. Detailed identification and characterization of such genes in lampreys is important for understanding how developmental gene regulatory programs diversified and became elaborated during vertebrate evolution. Thus, this study is an important contribution to these efforts.

This work has been carried out with a thorough attention to detail, the findings are novel, and the manuscript is well written and presented. Notably, the in situ hybridization and sectioning data are of very high quality, and these are the strongest aspect of the paper. However, I have concerns about the phylogenetic interpretations, which I would like to see addressed before I can fully recommend publication of this work. As detailed below, all I suggest is that the authors modify the strength of some of their claims regarding the phylogenetic correspondence between certain lamprey and gnathostome Noggin genes, as well as adding text to the abstract to highlight their detailed characterization of Noggin expression in lamprey.

Major points:

- i. The phylogenetic tree in Fig. 1A has a high bootstrap value (100) for the grouping of NogginD with Noggin4, which implies orthology between these proteins. This is fine. However, the relationships between the other lamprey Noggin proteins and their gnathostome homologues seem much less clear. For instance, the clade consisting of NogginA and Noggin1 has a low bootstrap value (18). Similarly, the clade consisting of NogginB, NogginC and Noggin2 also has a low bootstrap value (18). Thus, it appears that the relationships between NogginA,B,C and Noggin1,2 are not resolved with any clarity.

Furthermore, it is unclear how the local synteny analysis offers any further resolution, besides supporting the grouping of NogginA,B,C with Noggin1,2 and grouping NogginD with Noggin4. Therefore, the claim made in the discussion that ‘NogginA corresponds to gnathostome Noggin1, NogginB and NogginC correspond to Noggin2’ is not justified and should be toned down.

This lack of phylogenetic clarity is not surprising, given the divergence between lamprey and gnathostomes. I do not have a problem with the authors’ phylogenetic approach, just with their interpretation of the tree. I suggest the authors modify the strength of their claims regarding direct correspondence between NogginA and Noggin1, and NogginB,C and Noggin2.

Our answer:

We totally agree that it is impossible to make confident conclusions only on the basis of a tree with such bootstraps. In our opinion, the observed phylogeny looks more like a “cluster” or “cloud” homology.

We changed the text of this section:

“As we also revealed, by Maximum Likelihood (ML) protein analysis, NogginA appeared to be the closer to gnathostome Noggin1, while NogginB and NogginC appeared to be closer to Noggin2. However, as the bootstrap value justifying such clustering is quite low (<50%) in our opinion, it would be more correct to speak of cluster or cloud homology of the lamprey NogginA/B/C proteins on the one hand and jawed vertebrates’ Noggin1/2 on the other. (Figure 1A).”

and after the synteny analysis:

“This conclusion is consistent with the results of a clustering analysis that slightly gravitates NogginA to Noggin1 in one cluster and NogginB, NogginC to Noggin2 in another cluster.”

To modify the strength of our claims regarding *Noggins* phylogeny, we replaced term “corresponds” to “gravitates” in the Results and Discussion sections.

On our opinion, the additional support of presented phylogenetic view could come from expression patterns of *Noggins*, as the pattern lamprey *NogginA* has many similarities with jawed *Noggin1* and *NogginB* has the same specific pattern as *Noggin2*. Of course, the possibility of independent origin of so similar patterns in different lineages cannot be completely rejected, but looks unlikely, in our opinion.

We added these considerations in the updated Discussion version.

Reviewer #1

The discussion of the different possible scenarios leading to the complement of lamprey and gnathostome *Noggin* genes is speculative but suitably balanced, so does not require alteration.

Our answer:

We are grateful for your assessment of our Discussion section, but we updated it in accordance with the remarks of other reviewers.

Reviewer #1

ii. The authors do a great job characterizing the expression domains of the lamprey Noggin genes during embryogenesis and comparing these with their gnathostome counterparts, but this is not mentioned in the abstract. These are valuable results and should be referred to in the abstract as a major focus of the paper.

Our answer:

Thanks again for your kind words about our study.

We added in the Abstract:

“In this work, we described and investigated phylogeny and expression patterns of four Noggin genes in lampreys...”

Reviewer #2

This manuscript reported four Noggin homolog genes for the first time from lamprey (I don't think so. See below). They conducted phylogenetic tree analysis and genome synteny analysis, qPCR and in situ hybridization on the lamprey embryos. Finally, they did injection experiment by using *Xenopus* embryo whether the lamprey noggin genes can induce secondary body axes. Their main conclusion of this manuscript is that the latest common ancestor of jawless and jawed vertebrates has experienced two round whole genome duplications. This conclusion mainly comes from the gene phylogenetic analysis and gene synteny analysis.

Major comments:

Regarding identification of lamprey Noggin genes and phylogenetic analysis, it has been already published (Kuraku and Kuratani GBE 2011), which is not cited in this manuscript. Kuraku and Kuratani already identified four Noggin genes from sea lamprey genome and conducted phylogenetic tree analysis. In addition, the estimation of the Noggin gene family evolution is different. Especially, Kuraku concluded that *Noggin4/D* was split BEFORE the 2 round whole genome duplication. The authors should cite this paper and clarify where is the new point of the current study, and where the different conclusion came from.

Our answer:

Thanks for the reminder of the Kuraku and Kuratani article, we admit unfair the lack of reference to this article in our work.

In the same time, Kuraku and Kuratani paper was focused on genome-wide analysis of the genes lost in the early mammalian evolution, lamprey *Noggins* were included in the phylogenetic tree (as predicted sequences) but were not described at all. There is no analysis and/or at least a mention in the text of the article, since the focus is on the analysis of the previously described jawed vertebrate *Noggin* genes. At the same time, in the given tree, lamprey *Noggins* do not fall into the groups designated as *Noggin1*, *Noggin2* and even *Noggin4* and, probably, for this reason, they are named just as “*sea lamprey predicted*”. In this article there is neither a description of synteny, nor spatial/temporal expression features.

In our study we present *Noggin* genes in three lamprey species, analyze their synteny, expression in details and show functional properties.

Based on the data presented in Kuraku and Kuratani paper, their statement “This gene family experienced a gene duplication early in metazoan evolution between the *Noggin1/2/3* and *Noggin4* subfamilies, each of which are represented by the homologs of the invertebrates, such as the starlet sea anemone, *Nematostella vectensis*, and the freshwater planarian, *Schmidtea mediterranea* ” is made essentially only on the basis of a phylogenetic tree.

Not excluding this possibility, we propose to comprehensively consider other possible scenarios in the last part of the updated Discussion section, expressing our opinion on their probability. To be short, the main question to the early split of *Noggin1/2/3* and *Noggin4* is the absence of *Noggin4* in all invertebrate chordates.

We cited Kuraku and Kuratani paper and changed the text in the Introduction:

*“The fact of the presence in cyclostomes of several Noggin genes was noticed for the first time for the sea lamprey *Petromyzon marinus*²⁵. However, since this work was focused on the genome-wide analysis of genes lost in the early mammalian evolution, these lamprey Noggins, while been included in the common phylogenetic analysis, were not described and investigated in details. In the present work, we revealed that lampreys have four Noggin genes and then studied their phylogeny, local genomic synteny, expression patterns and the ability to induce secondary body axes, including the head. Establishing that lampreys have orthologues of *Noggin1*, *Noggin2* and *Noggin4* in gnathostomes, we consider their possible evolution in the context of the existing models of genome duplications in the early vertebrate history. The obtained results incline us to the hypothesis suggesting two rounds of genome duplication in the common ancestor of vertebrates before divergence of jawless and gnathostomes.”*

Also we removed the sentences that we describes lamprey genes “for the first time” from the Abstract and Discussion.

Reviewer #2

Gene expression analysis provides new insights on the lamprey Noggin genes. Only minor anatomical corrections are needed (see below).

The injection experiments are clear and quite convincing. However, it is already reported that Noggin genes from more distant invertebrate animals (hydra and planaria) have induction activity for the second axis as they cited (refs. 27, 28). Therefore, it is not so much valuable information from the evolutionary point of view. In addition, this experiment is much less related to the main conclusion.

Our answer:

We do not claim here, that the ability of Noggin to induce additional body axes is the new knowledge (although this property has not been shown previously for lamprey Noggins). We just use this is a conservative Noggins property as a convenient and descriptive test to evaluate the functionality of the genes studied.

Cited Molina et al. and Chandramore et al. articles demonstrate the ability of planarian and hydra Noggins to induce secondary axes, but not head structures.

“Overexpression of both Noggin mRNAs induces the incomplete secondary axes (arrow)” (Chandramore K et al., 2010).

As we showed already after the publication of these articles in Bayramov et al., 2011 (Development), the ability to induce additional head structures is an important functional property of vertebrate *Noggin1/2* genes.

Therefore, one of the central aspects of our current article is an analysis of the expression of lamprey Noggins in brain structures that could reflect their possible involvement in the

development of forebrain structures, especially telencephalon, that appears for the first time in the evolution just in Cyclostomes.

Reviewer #2

Minor comments:

L39: The extant cyclostome is NOT the closest to ancestral vertebrates than modern gnathostomes. Because both groups are equally evolved from the latest common ancestors. Extant cyclostomes have many specific traits, which acquired through their lineages.

Our answer:

It's absolutely fair remark. We changed this sentence for:

“These data allow one to consider jawless, including extant cyclostomes, lampreys and hagfishes, as the most basally divergent group of vertebrates.”

Reviewer #2

L126, L149: Please clarify which amino acid substitutions are critical for the BMP binding activity. Also, please provide us whether those substitutions exist or not in the Nodal4/D.

Our answer:

Noggin4/D have following substitutions: P to L at p.56; E to S at p.69; I to R at p.288. We marked them at the Alignment (Supplementary, Figure 1) and added the link to the text.

“Moreover, NogginD, the lamprey orthologue of Noggin4, had amino acid substitutions in positions critical for the binding of BMP, which presumably would prevent its binding with these ligands (Supplementary, Figure 1)^{20,23}.”

The importance of these positions for BMP binding was described in (Groppe et al., 2002, Nesterenko et al., 2016).

Reviewer #2

L140, L185, L198 etc.: Please remove the underlines.

Our answer:

Done.

Reviewer #2

L199, L243: intERthalamica -> intRAthalamica

Our answer:

Fixed.

Reviewer #2

Fig.4E, Fig5N: The position of “mes” is wrong. Mesencephalon should be positioned anterior to the MHB.

Our answer:

Thanks for the amendment. Fixed.

Reviewer #2

There are some typos, Russian character and wrong use of italic characters (i.g. L227 “NogginC is expressed”). Please correct those mistakes carefully or use proofreading services.

Our answer:

We fixed these technical points and reviewed the text by English language Editing in SpringerNatureAuthor service (certificate may be verified on the SNAS website using the verification code 1BEA-1044-88EB-B82A-5242).

Reviewer #3:

The manuscript “Discovery of four Noggin genes in lampreys may suggest two rounds of genome duplication preceding divergence of Agnatha and Gnathostomata” by Ermakova et al makes a compelling case that the noggin gene underwent two rounds of duplication and evolved specific functions before the divergence of vertebrate and lamprey lineages. Other reviewers can likely provide more accurate comments on the details of expression, so I will limit mine to the discussion.

My only major comments are related to necessary modification to the introduction and discussions regarding duplication scenarios. These should be carefully revised to reflect the referenced studies and the data that are available through this study.

1) An important part of the published 1R model is that some/much of the signal that has been traditionally interpreted as supporting a gnathostome 2R is from local duplications that pre-dated (or postdated) a shared whole duplication event, with no subsequent whole genome duplications in the gnathostome lineage. The data presented show limited conserved synteny in the vicinity of noggin, while this supports two rounds of duplication, it cannot differentiate local vs whole genome duplication. Therefore a statement like “This model is in good agreement with the hypothesis suggesting two rounds of whole-genome duplication in the common ancestor of vertebrates before divergence of jawless and gnathostomes.” Should be reworded to “This model is in good agreement with the hypothesis suggesting two rounds of duplication in the common ancestor of vertebrates before divergence of jawless and gnathostomes.” OR “This model is in good agreement with the hypothesis suggesting two rounds of local and/or whole genome duplication in the common ancestor of vertebrates before divergence of jawless and gnathostomes.”

Our answer:

Absolutely fair remark. We changed the sentence in the introduction:

“The obtained results incline us to the hypothesis suggesting two rounds of genome duplication in the common ancestor of vertebrates before divergence of jawless and gnathostomes.”

Reviewer #3:

2) Incorporating this will require careful modification of the discussion section.

Our answer:

We updated Discussion section and divided it now on three parts. First and second summarize the data obtained about lamprey *Noggin*s and the third one discusses possible scenarios of emergence vertebrates *Noggin* family in evolution.

Now we discuss in details and present as scheme two possible scenarios, based on the analysis of phylogeny, synteny, and expression patterns of vertebrate *Noggin*s, trying to note the convincing sides and compelled assumptions of each of scenarios.

We would like to emphasize, that our goal here in is not to prove the truth of any of the scenarios presented, since this looks unreal with the example of only one family of genes. Rather, we are trying to correlate our data obtained with those models that are being discussed today and express our opinion on which scenario seems more likely to us.

The part of the Discussion section looks like that now:

“The hypothesis of two rounds of whole genome duplications before divergence of jawless fish and gnathostomes may explain the presence of the discovered Noggin genes in these two branches of extant vertebrates

As is currently discussed in the scientific community, this initial period of vertebrate evolution included either one or two rounds of whole ancestral genome duplication^{5, 38}. Initially, this supposition was inspired by the finding of four and six complexes of HOX genes in gnathostomes and lampreys, respectively^{39, 40, 41}. The nearest relatives of these vertebrates, tunicates and cephalochordates, have only one HOX complex. Thereafter, similar results were obtained for the Pax6 gene⁴². These findings indicated that either two rounds of genome duplication occurred before the divergence of the jawless fish and gnathostomes, followed by an additional round of the entire or partial genomic duplications in the jawless fish, or that only one round of the duplication preceded the divergence of the jawless fish and gnathostomes, with additional rounds occurring independently in these two branches of vertebrates^{4, 5, 43}. Accordingly, in order to judge which of these two scenarios is more realistic, one may try to analyse which of them is more consistent with the appearance of the Noggin genes that have now been found in lampreys and gnathostomes.

The revealed phylogeny, synteny, expression patterns and functional analyses show that vertebrates Noggin can be confidently divided into two clusters: NogginA/B/C/1/2 and NogginD/4. At the same time, the expression pattern of lamprey NogginA has many similarities with jawed Noggin1, while NogginB has the expression pattern very similar to Noggin2. As the independent origin of so similar patterns in different lineages looks in our opinion very unlikely, one may hypothesize that the common vertebrate ancestor should have had at least three Noggin genes before split of cyclostomes and gnathostomes. In turn, this could be possible only in two cases: (a) one “basal” Noggin gene in vertebrate ancestor passed through two rounds of whole-genome duplications before the divergence of the jawless fishes and gnathostomes; (b) two

“basal” genes (obviously *NogginA/B/C/1/2* and *NogginD/4*) passed through one round of duplication in the common ancestor. Let us consider both these models.

If two rounds of whole-genome duplications preceded the divergence of the jawless fishes and gnathostomes, one may hypothesize that, during the first round, two copies of ancestral *Noggin1* arose, followed by the evolution of one of these copies into the close homologue of ancestral *Noggin1*: the ancestral *NogginB/NogginC/Noggin2* (Figure 7 A). During this time, the latter gene might have inherited at least two 5' genomic neighbours of the ancestral *Noggin1* (*MMD* and *ANKFN1*), which are still present in the vicinity of *NogginC*. Then, two copies of ancestral *Noggin1* and *NogginB/NogginC/Noggin2* could have appeared as a result of the second round of whole genome duplication. During this time or later, one copy of ancestral *NogginB/NogginC/Noggin2* could have lost the homologue of *MMD*, thus acquiring the specific 3' region surrounding the evolved *NogginB* and *Noggin2* genes (the ancestor of *NogginB/Noggin2*). At the same time, the second copy of ancestral *NogginB/NogginC/Noggin2*, which maintained both *MMD* and *ANKFN1* near the 5' end, became the ancestor of *NogginC*. Thereafter, one of two copies of ancestral *Noggin1* might have been translocated to a different genomic location, followed by specific point mutations in this translocated copy, which impaired the ability of its protein product to bind BMP. Finally, further mutagenesis could have been generated from this translocated copy of ancestral *Noggin1*, which is the common ancestor of *NogginD* and *Noggin4*. After the subsequent separation of the jawless and gnathostomes, one copy of ancestral *Noggin1*, the ancestor of *NogginB/Noggin2* and *NogginC*, as well as the ancestor of *Noggin4*, could have been inherited by extant lampreys as *NogginA*, *B*, *C* and *D*. At the same time, in the vertebrate branch, the ancestor of *NogginC* seemingly was lost, resulting in only three *Noggin* genes now present in most of the extant vertebrates: *Noggin1*, *Noggin2* and *Noggin4*.

In case of the one-round scenario of whole-genome duplication, one has to allow the presence of two *Noggin* genes in vertebrate ancestor (Figure 7 B). After the subsequent single round of genomic duplication, vertebrates could obtain *NogginA/1*, *NogginB/C/2* and two copies of *Noggin4/D* genes. As all extant vertebrates have only one copy of *NogginD* or *Noggin4* gene, another copy of this ancestor *Noggin4/D* had to disappear somewhere in-between the moment of the whole genome duplication and the moment of the divergence of jawless and gnathostomes. The appearance of *NogginB* and *NogginC* in this model could be the result of a local duplication and further divergence of the descendants of *NogginB/C/2* in lampreys.

In support of the hypothesis of the existence of two “basic” *Noggin*s in vertebrate ancestor one may point on the presence of multiple *noggin*-like genes (*nlg*) in the flat worm *Schmidtea mediterranea*^{28, 44}, which could be considered as possible predecessors of the hypothetical *NogginA/B/C/1/2* and *NogginD/4* in the vertebrate ancestor. As was shown by the authors using *X. laevis* embryos as models, at least some of the *nlg*s were indeed able to bind BMP, like vertebrate *Noggin1* and 2, while other not, like *Noggin4*.

However, despite we cannot absolutely exclude the possibility of such a scenario, on our opinion, it is very difficult to imagine direct phylogenetic relationship of the flat worm *nlg*s with vertebrate *Noggin4* because of the aforementioned lack of *Noggin4* homologs genes in all extant deuterostome predecessors of vertebrates that separate them from flat worms. Therefore, it seems more logical to consider that these *nlg*s emerged independently of vertebrate *Noggin4* during the evolution of the flat worm branch. Such independent appearance of *Noggin4* and *noggin*-like genes, for which the protein products lack the ability to bind BMP, could be based simply on the emergence of multiple copies of *Noggin* genes in any animal lineage. In this case, one copy of *Noggin* may continue to retain the ability to bind BMP, while at the expense of this feature, another copy(s) escaped the influence of stabilizing natural selection and quickly acquired point mutations in any one of the four amino acid positions, which have been shown to be critical for BMP binding²³. At the same time, the mutant copy of *Noggin* may not have been eliminated during evolution because, as we previously showed, *Noggin* proteins, in addition to BMP binding, can bind ligands of other signalling pathways, such as those of the Wnt and

Activin/Nodal pathways, thereby participating in the regulation of other important developmental processes^{18, 20}.

Another possibility of the appearance of two basic Noggins in vertebrate ancestors could be if the ancestry *NogginD/4* gene would appear somewhere at the very beginning of the vertebrate evolution due to some local duplication of *NogginA/B/C/1/2*, followed by translocation to another genomic surrounding and further evolution of one of the duplicated copies into *NogginD/4*. However, no signs of the existence of such ancestry homologs of *NogginD* or *Noggin4* are revealed in any of the nearest extant relatives of vertebrates, including Cephalochordates, Tunicates or hemichordates.

Given all these considerations and being guided by Occam's razor principle, we believe that at the present time the hypothesis of two rounds of whole-genome duplications before the divergence of the jawless fish and gnathostomes may explain less controversially the presence of those Noggins that were revealed in lampreys and gnathostomes.

Figure 7. Two possible scenarios of evolution of *Noggin* genes in vertebrates.”

Reviewer #3:

3) Regarding this model, it will be important to reiterate that the authors are proposing that the duplications occurred AND the regulatory/functional changes evolved prior to divergence.

Our answer:

Right, according to our model, the duplication of *Noggin* genes and a change in their functional properties occurred before the divergence of the lampreys and gnathostomes.

One of the arguments here is conservatism of expression patterns and functional properties between *NogginA/1*, *NogginB/2* and *NogginD/4* in lampreys and gnathostomes and the likelihood of the independent appearance of such parallelism in different lineages. This scenario cannot be completely excluded, but it looks less likely, on our opinion.

At the same time, the main question to the idea of two basic ancestor *Noggins* (i.e. *Noggin1/2* and *Noggin4*) in all metazoan is the lack of *Noggin4* in all invertebrate Chordates.

Reviewer #3:

4) The findings presented here would seem to be inconsistent with the recently proposed 2R model (PMID: 32313176) that was originally presented as alternative to explain 1R(plus local)-like patterns

Our answer:

Thanks a lot for the link for the new model, we obviously couldn't see it before as it was published April 20th.

For our case such a model, which implies one round of WGD in the ancestor of all vertebrates and one subsequent duplication in gnathostomes, has a lot in common with the 1R model, but moreover raises questions about additional *Noggin* copies in gnathostomes.

Reviewer #3:

Minor comment:

Use of the term “cyclostome” invokes a specific evolutionary model wherein hagfish and lamprey are considered sister taxa, however it should be noted that phylogenetic support for this hypothesis is weak at best (see PMID: 25071211) the authors might reconsider use of the term, especially since they understandably do not include hagfish data presumably due to the fragmentary nature of the available assembly and difficulty in accessing embryos. Cyclostome is a widely used term the definition of which has little bearing on this study, so it may not be absolutely necessary to dwell on that for the purposes of this manuscript.

Sincerely,
Jeramiah Smith

Our answer:

As the analysis of current views on the evolutionary relationships of lampreys, hagfishes and gnathostomes is beyond the matter of our article, we use “cyclostomes” just as widely used “classical” term.

According to the wishes of the reviewer we removed or replaced “cyclostomes” to “lampreys” or “representatives of cyclostomes” at L. 38, 46, 548.

We left “cyclostomes” mostly in the theoretical parts - Abstract, Introduction and in the last part of Discussion where the use of this term could be relevant, on our opinion.

Reviewers' comments:

Reviewer #1 (Remarks to the Author):

I am satisfied by the authors' responses and modifications to the manuscript in response to my previous concerns. I have no further concerns or suggestions for modifications.

Reviewer #3 (Remarks to the Author):

The authors seem to have missed one of the primary points raised in my first review. In their revision they present an alternate scenario that would seem to be substantially different from any proposed to date - laid out in panel B of a version of figure 7 that was attached to the reply. This unnecessarily complicated scenario would certainly be less favored by applying the principles of Occam's razor, however, at least one other scenario raised in the previous review would not. Specifically, if the duplication that is attributed to a first round of whole genome duplication in Figure 7 A were replaced by a local duplication of the gene and two flanking genes it would yield a pattern that is indistinguishable from 2 rounds of whole genome duplication and which would perfectly fit both the classical 2R AND proposed alternatives.

An important point is that the author's work is less consistent with the specific details of at least one 2R-WGD reconstruction (PMID: 32313176) and likely others, under which the tuning of orthologous functions described in this paper seemingly could not have occurred prior to the split of lamprey and gnathostome lineages because the duplications necessarily post-date the split of these lineages.

The limitations of this study in defining the precise mode of duplication are seemingly easily resolved by replacing various instances of "whole genome duplication" with "duplication". These do not lessen the impact of the work and substantially clarify its relevance to the broader fields of research.

For example: "Discovery of four Noggin genes in lampreys may suggest two rounds of genome duplication preceding divergence of Agnatha and Gnathostomata"  "Discovery of four Noggin genes in lampreys suggests two rounds of duplication preceding divergence of Agnatha and Gnathostomata" (note that "may" and "suggest" are syntactically redundant)

and "This model is in agreement with the hypothesis suggesting two rounds of whole-genome duplication"  "This model is in agreement with hypotheses suggesting two rounds of whole-genome and or local duplication in the ancestor of vertebrates before the divergence of Agnatha and Gnathostomata." OR "This model is in agreement with hypotheses suggesting two rounds of duplication in the ancestor of vertebrates before the divergence of Agnatha and Gnathostomata."

We appreciate Editor's and Reviewers' comments.
Our detailed responses are below.

Editor's comment.

Reviewer 1 is satisfied by this revision, however still some important points remain that we ask to be addressed. Separately to the editor, a reviewer raised a concern that the relatively few samples of invertebrates included in the phylogenetic tree could influence the results, and we ask that you address this concern and include more outgroups in the analyses. We also ask that you address the alternative scenarios raised by Reviewer 3 and acknowledge the recent work of Simakov et al.

Our answer:

We initially included in our phylogenetic tree the representatives of the closest relatives of vertebrates - the invertebrate chordates - lancelet and ascidia, in which one Noggin gene was detected. The main question we posed when constructing this tree is the search for homologies between the Noggin genes in Cyclostomes vs Gnathostomes lineages. Since in basic chordates (lancelet and ascidia) only one Noggin gene was found, the origin of multiple Noggin genes in vertebrates and invertebrates groups is most likely independent, as discussed in our Discussion section. In Molina et al., 2009 article, that we cited, 2 planarian Noggins and 8 Noggin-like-genes are grouped closer to each other than to vertebrates' Noggin1 and Noggin2 (Molina et al., 2009, Fig.1). In addition, the phylogenetic tree of Noggin proteins, which includes many representatives of invertebrates, was presented in the article of Kuraku and Kuratani, 2011 (Fig.6) that we also cited. The main focus of their article is different from our, but it cannot be said that the phylogenetic analysis presented by the authors, that includes a significant number of invertebrate genes, shed the light to the understanding of the orthology of lampreys and Gnathostomes Noggins.

Anyway, following the Reviewer considerations and your recommendations we now add Noggin proteins of the following Invertebrates: *Ciona savygnyi*, *Asterias rubens*, *Anneissia japonica* and *Strongylocentrotus purpuratus* to our phylogenetic tree presented at Supplementary Figure 2. In addition to ML, we also presented NJ tree there.

Reviewer #3:

The authors seem to have missed one of the primary points raised in my first review. In their revision they present an alternate scenario that would seem to be substantially different from any proposed to date - laid out in panel B of a version of figure 7 that was attached to the reply. This unnecessarily complicated scenario would certainly be less favored by applying the principles of Occam's razor, however, at least one other scenario raised in the previous review would not. Specifically, if the duplication that is attributed to a first round of whole genome duplication in Figure 7 A were replaced by a local duplication of the gene and two flanking genes it would yield a pattern that is indistinguishable from 2 rounds of whole genome duplication and which would perfectly fit both the classical 2R AND proposed alternatives.

An important point is that the author's work is less consistent with the specific details of at least one 2R-WGD reconstruction (PMID: 32313176) and likely others, under which the tuning of orthologous functions described in this paper seemingly could not have occurred prior to the split of lamprey and gnathostome lineages because the duplications necessarily post-date the split of

these lineages.

The limitations of this study in defining the precise mode of duplication are seemingly easily resolved by replacing various instances of “whole genome duplication” with “duplication”. These do not lessen the impact of the work and substantially clarify its relevance to the broader fields of research.

For example: “Discovery of four *Noggin* genes in lampreys may suggest two rounds of genome duplication preceding divergence of Agnatha and Gnathostomata”  “Discovery of four *Noggin* genes in lampreys suggests two rounds of duplication preceding divergence of Agnatha and Gnathostomata” (note that “may” and “suggest” are syntactically redundant)

and “This model is in agreement with the hypothesis suggesting two rounds of whole-genome duplication”  “This model is in agreement with hypotheses suggesting two rounds of whole-genome and or local duplication in the ancestor of vertebrates before the divergence of Agnatha and Gnathostomata.” OR “This model is in agreement with hypotheses suggesting two rounds of duplication in the ancestor of vertebrates before the divergence of Agnatha and Gnathostomata.”

Our answer:

We apologize to the Reviewer for we did not understand his basic idea in the first review. Yes, we agree that, based on our data, it is difficult to judge for sure whether the first round of duplication was whole-genome or local. Therefore, in the revised version, we also added scheme of the scenario with 1WGD + local as a possible alternative to 2WGD and replaced “whole genome duplication” with “genome duplication” in the text.

Thanks again for the clarifications and suggestions!

The last section of the discussion now looks as the following:

“The hypotheses suggesting two rounds of genome duplications before divergence of jawless fish and gnathostomes may explain the presence of the discovered *Noggin* genes in these two branches of extant vertebrates

As is currently discussed in the scientific community, this initial period of vertebrate evolution included either one or two rounds of ancestral whole-genome duplication^{5, 38, 39}. Initially, this supposition was inspired by the finding of four and six complexes of *HOX* genes in gnathostomes and lampreys, respectively^{40, 41, 42}. The nearest relatives of these vertebrates, tunicates and cephalochordates, have only one *HOX* complex. Thereafter, similar results were obtained for the *Pax6* gene⁴³. These findings indicated that either two rounds of genome duplication occurred before the divergence of the jawless fish and gnathostomes, followed by an additional round of the entire or partial genomic duplications in the jawless fish, or that only one round of the duplication preceded the divergence of the jawless fish and gnathostomes, with additional rounds occurring independently in these two branches of vertebrates^{4, 5, 44}. Accordingly, in order to judge which of these two scenarios is more realistic, one may try to analyse which of them is more consistent with the appearance of the *Noggin* genes that have now been found in lampreys and gnathostomes.

The revealed phylogeny, synteny, expression patterns and functional analyses show that vertebrates *Noggins* can be confidently divided into two clusters: *NogginA/B/C/1/2* and *NogginD/4*. At the same time, the expression pattern of lamprey *NogginA* has many similarities with jawed *Noggin1*, while *NogginB* has the expression pattern very similar to *Noggin2*. As the independent origin of so similar patterns in different lineages looks in our opinion very unlikely, one may hypothesize that the common vertebrate ancestor should have had at least three *Noggin* genes before split of cyclostomes and gnathostomes. **Based on these**

considerations, our data are not consistent with the recently proposed model suggesting only one common duplication before split of Cyclostomes and Gnathostomes followed by additional independent duplications after separation of these lineages⁴⁵. Appearance of at least three *Noggins* in Cyclostomes and Gnathostomes common ancestor could be possible only in two cases: (a) one "basal" *Noggin* gene in vertebrate ancestor passed through two rounds of genome duplications before the divergence of the jawless fishes and gnathostomes; (b) two "basal" genes (obviously *NogginA/B/C/1/2* and *NogginD/4*) passed through one round of duplication in the common ancestor. Let us consider both these models.

If two rounds of whole-genome duplications preceded the divergence of the jawless fishes and gnathostomes, one may hypothesize that, during the first round, two copies of ancestral *Noggin1* arose, followed by the evolution of one of these copies into the close homologue of ancestral *Noggin1*: the ancestral *NogginB/NogginC/Noggin2* (Figure 7 A). During this time, the latter gene might have inherited at least two 5' genomic neighbours of the ancestral *Noggin1* (*MMD* and *ANKFN1*), which are still present in the vicinity of *NogginC*. Then, two copies of ancestral *Noggin1* and *NogginB/NogginC/Noggin2* could have appeared as a result of the second round of whole genome duplication. During this time or later, one copy of ancestral *NogginB/NogginC/Noggin2* could have lost the homologue of *MMD*, thus acquiring the specific 3' region surrounding the evolved *NogginB* and *Noggin2* genes (the ancestor of *NogginB/Noggin2*). At the same time, the second copy of ancestral *NogginB/NogginC/Noggin2*, which maintained both *MMD* and *ANKFN1* near the 5' end, became the ancestor of *NogginC*. Thereafter, one of two copies of ancestral *Noggin1* might have been translocated to a different genomic location, followed by specific point mutations in this translocated copy, which impaired the ability of its protein product to bind BMP. Finally, further mutagenesis could have been generated from this translocated copy of ancestral *Noggin1*, which is the common ancestor of *NogginD* and *Noggin4*. After the subsequent separation of the jawless and gnathostomes, one copy of ancestral *Noggin1*, the ancestor of *NogginB/Noggin2* and *NogginC*, as well as the ancestor of *Noggin4*, could have been inherited by extant lampreys as *NogginA*, *B*, *C* and *D*. At the same time, in the vertebrate branch, the ancestor of *NogginC* seemingly was lost, resulting in only three *Noggin* genes now present in most of the extant vertebrates: *Noggin1*, *Noggin2* and *Noggin4*.

The possible alternative of this scenario could include local duplication of ancestor *Noggin* gene along with its adjacent genes instead the whole-genome duplication at the first step (Figure 7B). However, basing on the data of the present work, it is difficult to judge whether the first round of duplication was whole-genome or local.

In case of the one-round scenario of whole-genome duplication, one has to allow the presence of two *Noggin* genes in vertebrate ancestor (Figure 7 C). After the subsequent single round of genomic duplication, vertebrates could obtain *NogginA/1*, *NogginB/C/2* and two copies of *Noggin4/D* genes. As all extant vertebrates have only one copy of *NogginD* or *Noggin4* gene, another copy of this ancestor *Noggin4/D* had to disappear somewhere in-between the moment of the whole-genome duplication and the moment of the divergence of jawless and gnathostomes. The appearance of *NogginB* and *NogginC* in this model could be the result of a local duplication and further divergence of the descendants of *NogginB/C/2* in lampreys.

In support of the hypothesis of the existence of two "basic" *Noggins* in vertebrate ancestor one may point on the presence of multiple *noggin*-like genes (*nlg*) in the flat worm *Schmidtea mediterranea*^{28, 46}, which could be considered as possible predecessors of the hypothetical *NogginA/B/C/1/2* and *NogginD/4* in the vertebrate ancestor. As was shown by the authors using *X. laevis* embryos as models, at least some of the *nlg*s were indeed able to bind BMP, like vertebrate *Noggin1* and 2, while other not, like *Noggin4*.

However, despite we cannot absolutely exclude the possibility of such a scenario, on our opinion, it is very difficult to imagine direct phylogenetic relationship of the flat worm *nlg*s with vertebrate *Noggin4* because of the aforementioned lack of *Noggin4* homologs genes in all

extant deuterostome predecessors of vertebrates that separate them from flat worms. Therefore, it seems more logical to consider that these *nlg*s emerged independently of vertebrate *Noggin4* during the evolution of the flat worm branch. Such independent appearance of *Noggin4* and *noggin*-like genes, for which the protein products lack the ability to bind BMP, could be based simply on the emergence of multiple copies of *Noggin* genes in any animal lineage. In this case, one copy of *Noggin* may continue to retain the ability to bind BMP, while at the expense of this feature, another copy(s) escaped the influence of stabilizing natural selection and quickly acquired point mutations in any one of the four amino acid positions, which have been shown to be critical for BMP binding²³. At the same time, the mutant copy of *Noggin* may not have been eliminated during evolution because, as we previously showed, *Noggin* proteins, in addition to BMP binding, can bind ligands of other signalling pathways, such as those of the Wnt and Activin/Nodal pathways, thereby participating in the regulation of other important developmental processes^{18,20}.

Another possibility of the appearance of two basic *Noggins* in vertebrate ancestors could be if the ancestry *NogginD/4* gene would appear somewhere at the very beginning of the vertebrate evolution due to some local duplication of *NogginA/B/C/1/2*, followed by translocation to another genomic surrounding and further evolution of one of the duplicated copies into *NogginD/4*. However, no signs of the existence of such ancestry homologs of *NogginD* or *Noggin4* are revealed in any of the nearest extant relatives of vertebrates, including Cephalochordates, Tunicates and Hemichordates.

Given all these considerations and being guided by Occam's razor principle, we believe that at the present time the hypothesis of two rounds of genome duplications before the divergence of the jawless fish and gnathostomes may explain less controversially the presence of those *Noggins* that were revealed in lampreys and gnathostomes.

Figure 7. Three possible scenarios of the evolution of *Noggin* genes in vertebrates.“

We also cited the article by Simakov et al., 2020 with considerations that, in our opinion, such a model is not entirely consistent with our data.

“Based on these considerations, our data are not consistent with the recently proposed model suggesting only one common duplication before split of Cyclostomes and Gnathostomes followed by additional independent duplications after separation of these lineages⁴⁵. Appearance of at least three Noggins in Cyclostomes and Gnathostomes common ancestor could be possible only in two cases... etc”.

REVIEWERS' COMMENTS:

Reviewer #3 (Remarks to the Author):

The alterations to the text seem appropriate, I only have a couple of comments regarding integration and grammar in the new sections.

1) "Appearance of at least three Noggins in Cyclostomes and Gnathostomes common ancestor could be possible only in two cases: (a) one "basal" Noggin gene in vertebrate ancestor passed through two rounds of genome duplications" ... This could be bit more precise regarding the use of functional evidence, and more careful about the number of possible explanations, particularly given that there are three presented in figure 7. I would suggest something like (Changes in BOLD CAPS or strikethrough in the attached document) "Appearance of at least three common functionally diverged Noggins in the cyclostome and gnathostome common ancestor is consistent with two general scenarios: (a) one "basal" Noggin gene in vertebrate ancestor passed through two rounds of duplication"

2) "The possible alternative of this scenario could include local duplication of ancestor Noggin gene along with its adjacent genes instead the whole-genome duplication at the first step (Figure 7B). However, basing on the data of the present work, it is difficult to judge whether the first round of duplication was whole-genome or local." This short addition could seemingly be joined to the preceding paragraph and edited for grammar like (Changes in BOLD CAPS or strikethrough in the attached document) "A possible alternative to this scenario could include local duplication of ancestor Noggin gene along with its adjacent genes instead of a whole-genome duplication at the first step (Figure 7B). Based on the present work, it is not possible to judge whether the first round of duplication was whole-genome or local."

3) Considering the added text and tension with the leading (Simkov) model and others, the authors might reconsider using the word "genome" in the title

We appreciate Reviewers' comments.
Our detailed responses are below.

Reviewer #3:

The alterations to the text seem appropriate, I only have a couple of comments regarding integration and grammar in the new sections.

1) "Appearance of at least three Noggins in Cyclostomes and Gnathostomes common ancestor could be possible only in two cases: (a) one "basal" Noggin gene in vertebrate ancestor passed through two rounds of genome duplications" ... This could be bit more precise regarding the use of functional evidence, and more careful about the number of possible explanations, particularly given that there are three presented in figure 7. I would suggest something like (Changes in **BOLD CAPS** or strikethrough) "Appearance of at least three **COMMON FUNCTIONALLY DIVERGED** Noggins in **THE ~~C~~cyclostomes and ~~G~~gnathostomes common ancestor IS CONSISTENT WITH** two **GENERAL SCENARIOS**: (a) one "basal" Noggin gene in vertebrate ancestor passed through two rounds of **genome** duplications"

Our answer:

We agree with this comment and we changed this sentence in the text:

"... appearance of at least three common functionally diverged Noggins in the cyclostome and gnathostome common ancestor is consistent with two general scenarios: (a) one "basal" Noggin gene in vertebrate ancestor passed through two rounds of duplication..."

Reviewer #3:

2) "The possible alternative of this scenario could include local duplication of ancestor Noggin gene along with its adjacent genes instead the whole-genome duplication at the first step (Figure 7B). However, basing on the data of the present work, it is difficult to judge whether the first round of duplication was whole-genome or local." This short addition could seemingly be joined to the preceding paragraph and edited for grammar like "A possible alternative to this scenario could include local duplication of ancestor Noggin gene along with its adjacent genes instead **OF A** whole-genome duplication at the first step (Figure 7B). **BASED** on the present work, it is **NOT POSSIBLE** to judge whether the first round of duplication was whole-genome or local."

Our answer:

We agree with this comment and we changed this sentence in the text:

"A possible variant of this scenario could include local duplication of ancestor *Noggin* gene along with its adjacent genes instead of a whole-genome duplication at the first step (Figure 7B). Based on the data of the present work, it is not possible to judge whether the first round of duplication was whole-genome or local."

Reviewer #3:

3) Considering the added text and tension with the leading (Simkov) model and others, the authors might reconsider using the word “genome” in the title

Our answer:

In our opinion, the use of the term “genome” in the title is permissible, as we use it in terms of “genetic information”, not necessarily a “whole genome”, as is discussed in the Discussion section.

To meet the requirements of the journal for the length of title we entitled the article according to the Editor recommendation: **“Discovery of four Noggin genes in lampreys suggests two rounds of ancient genome duplication”**.